# Olaparib, durvalumab, and cyclophosphamide, and a prognostic blood signature in platinum-sensitive ovarian cancer: the randomized phase 2 SOLACE2 trial

SOLACE2 (ACTRN12618000686202) investigates whether 12-weeks of olaparib, or cyclophosphamide-olaparib priming, improves subsequent durvalumab-olaparib progression-free survival (PFS), and is superior to olaparib monotherapy without any priming, in platinum-sensitive recurrent ovarian cancer (n = 114). We also evaluate the utility of CUP-CC assay, an immune signature of C-C chemokine receptor type 4 up-regulation, chemokines, and cytokines. Priming with olaparib, or cyclophosphamide-olaparib, followed by durvalumab-olaparib, are both associated with longer PFS compared to olaparib monotherapy, but do not reach the pre-specified primary endpoint of 36-week trial threshold (PFS36). PFS36 rates are 47.4% (95% CI, 31.0-62.1; olaparib priming then olaparib-durvalumab), 48.7% (32.5-63.2; olaparib-cyclophosphamide then olaparib-durvalumab) and 35.1% (20.4-50.3; olaparib monotherapy). PFS is significantly longer for the homologous recombination deficient (N = 71) as compared to the proficient (HRP) (N = 29) subgroups (Hazard Ratio (HR) 0.55, 0.35-0.87). CUP-CC+ subgroup (N = 58) has a significantly longer PFS (HR 0.31, 0.19-0.49) than CUP-CC- (N = 46). Future studies should investigate whether CUP-CC has the potential to personalize poly (ADP-ribose) polymerase inhibitor therapies for patients who are *BRCA* wild-type, including HRP patients.

Epithelial ovarian cancer (EOC) is the global leading cause of death from all gynecological cancers[1]. In 2020, there were more than 300,000 new cases globally, and more than 200,000 cancer–related deaths[2]. Most women have advanced-stage disease at presentation and are typically treated with surgery, platinum-taxane chemotherapy, and bevacizumab. Although these treatments are associated with high rates of response, the majority will still experience recurrence.

Platinum-sensitive recurrent high-grade serous ovarian cancer (PSROC) has conventionally been defined as cancer progression ≥6 months after the most recent platinum-based chemotherapy. There is a 50% likelihood of response with retreatment with platinum-based chemotherapy, but there is a declining likelihood of chemotherapy response with each successive line of treatment[3]. The addition of bevacizumab increases the response rate to 70% and prolongs progression-free survival (PFS)[4]. However, the median overall survival

e-mail: chee.lee@sydney.edu.au

(OS) is still less than 5 years, and is similar with chemotherapy alone or with the addition of bevacizumab.

Poly(ADP ribose) polymerase inhibitors (PARPi), either alone or in combination with bevacizumab, have regulatory approval as maintenance therapy following response to first-line platinum-based chemotherapy. The greatest benefits of PARPi are observed in patients with either germline or somatic *BRCA* pathogenic variants, and the least benefit in those with homologous recombination proficient (HRP) tumors. The PRIMA trial of maintenance niraparib in advanced stage high grade serous cancer (HGSOC) recently reported 5-year PFS rates of 22% with niraparib versus 12% with placebo in the intention-to-treat population, with PFS improvement for both HRP and homologous recombination deficient (HRD) subpopulations[5]. However, there was no difference in the median OS between treatment arms, including those with HRD and HRP tumors. The PAOLA-1 trial of first-line maintenance olaparib plus bevacizumab versus bevacizumab also reported no significant difference in OS in the intention-to-treat population, but 5-year OS rates in the HRD population were 65.5% and 48.4% in the olaparib plus bevacizumab and bevacizumab arms, respectively[6]. Finally, the addition of durvalumab during both carboplatin-paclitaxel chemotherapy (PC) and maintenance olaparib plus bevacizumab was investigated in the DUO-O trial[7] and showed a significant PFS prolongation in the *BRCA* wild-type with or without Myriad HRD positive populations, compared to PC-bevacizumab monotherapy, but OS data remain immature, and the comparator arm of PC-bevacizumab followed by maintenance bevacizumab plus olaparib was not included, hence preventing certainty around the contribution of durvalumab.

The impact of immune checkpoint inhibitors (ICI) in women with EOC has been modest[8,9]. EOC creates an immunosuppressive tumor microenvironment (TME) by preferentially chemoattracting CCR4-expressing regulatory T-cells (Treg)[8]. Treg in the TME employ multiple mechanisms to suppress CD8 T-cells: via expression of PD-L1 and CTLA-4; secretion of immunosuppressive cytokines IL-10, TGF-β, and IL-35; adenosine production by expression of ectoenzymes CD39 and CD73; and other mechanisms[10]. Tregs also modulate antigen-presenting cells (APCs) in the TME, impairing the ability of APCs to induce and expand CD8 T-cells. The use of ICI to interfere with any single immunosuppressive mechanism employed by Treg, is therefore unlikely to allow sufficient re-engagement of CD8 T-cell effector activity, to result in impressive anti-tumor efficacy. Low-dose cyclophosphamide treatment (LDCy) leads to Treg depletion and increased CD8 T cell/Treg ratios in the TME[11]. We have previously conducted a phase 1 study and showed that the combination of olaparib and LDCy was tolerable and had promising clinical activity, particularly in *BRCA* mutated EOC[12]. We therefore hypothesized that LDCy would help remodel the TME, priming the TME for optimal responsiveness to PARPi combination therapies.

PARPi therapy in EOC has been most effective in those with HRD, as DNA damage which arises from single-strand DNA breaks (SSBs) cannot be accurately repaired. In these cancers, PARPi exerts its therapeutic effects through the blockade of DNA damage repair, including of SSBs, leading to the accumulation of toxic DNA double-strand breaks[13]. Of relevance here, PARPi modulates the immune system through the stimulation of the interferon genes (STING) pathway. PARPi induces DNA damage activates the cGAS-STING pathway, increasing IFN-γ release and enhancing T-cell-dendritic cell crosstalk and promoting antigen presentation[14,15]. PARPi have synergistic effects with ICI, through DNA damage, cell death, and neoantigen and other antigen release, thereby enhancing ICI and intrinsic immune responses, including enhanced antigen presentation, increasing tumor-infiltrating lymphocytes, upregulation of PD-L1, and reprogramming of other molecules and immune cells involved in the TME[16,17]. In this context, it is plausible that prior PARPi alone may prime the TME for subsequent PARPi-ICI therapy. In addition, an increased CD8 T-cell to Treg ratio promoted by LDCy[18], could free the anti-tumor activity of the CD8 T-cells activated by PARPi treatment from Treg immunosuppression.

Studies have shown that high ratios of CD4 + Th1 to Treg or CD8 + T cell to Treg are strongly associated with decreased cancer recurrence[18,19]. Tregs are preferentially attracted into the TME as they constitutively express CCR4. This enables migration towards the cognate CCR4 chemokine ligands, CCL17 and CCL22, which are abundant in the TME. By contrast, Th1 and CD8 + T cells that normally express CCR5 and CXCR3, are important for immune surveillance, but their migration from blood into the TME is impaired because of destruction of CXCL10 by EOC cells[20].

Data presented here demonstrate that LDCy further modulates the expression of CCR4 on conventional CD4+ and CD8 + T cells. On this basis, we further hypothesized that LDCy would prime the ovarian TME, through enrichment with effector CD4+ and CD8 + T cells, with their function subsequently enhanced by ICI treatment, and we developed an assay to study CCR4 up-regulation (CUP). While high levels of IL-6, IL-8, CCL22, and CCL17 are all associated with poor prognosis[21–25], we speculated that individuals who responded by upregulating CCR4 on effector CD4+ and CD8 + T cells following mafosphamide challenge, and also had elevated levels of CCL22 and CCL17, would have improved prognosis. An immune signature, CUP-CC, which quantitatively combined CCR4 up-regulation, CCL22 and CCL17 levels, as well as IL-6 and IL-8, was, therefore, developed.

The SOLACE2 trial investigated two immune priming strategies, with either olaparib plus LDCy or olaparib monotherapy, followed by consolidation treatment with olaparib plus durvalumab to improve PFS in PSROC (Supplementary material of the SOLACE2 trial: Supplementary note 1, Supplementary Fig. 1). The chemotherapy-sparing interventions were offered as active therapies at the time of progression, rather than as maintenance treatment, for participants where chemotherapy was not yet indicated. We also concurrently and prospectively evaluated the prognostic potential of CUP-CC and whether it will also be predictive for additional PFS benefit with durvalumab.

In this work, we show that PFS was longer in participants who received priming olaparib, or cyclophosphamide-olaparib, followed by durvalumab-olaparib, compared to olaparib monotherapy, but this did not reach the pre-specified primary endpoint of a 36-week trial threshold. The CUP-CC positive subgroup had a significantly longer PFS compared to those who were CUP-CC negative, independent of HRD status.

## Results

From February 2019 to October 2022, 115 participants underwent random assignment, with one ineligible participant excluded post-randomization, who did not receive assigned therapy. Participants were assigned to receive olaparib followed by olaparib-durvalumab (arm A, N = 38), olaparib-LDCy followed by olaparib-durvalumab (arm B, N = 39), or olaparib monotherapy (arm C, N = 37). All participants in arms A, B, and C received the assigned study treatment during the priming phase. During the consolidation treatment phase, 37 (97.4%), 35 (89.7%), and 30 (81.1%), respectively, received assigned treatment (Fig. 1).

Baseline characteristics were generally balanced across treatment arms (Table 1) and representative of PSROC population.

During the priming phase, the relative dose intensity (RDI) of olaparib, defined as the ratio of the actual dose received to the protocol-mandated dose, was similar across arms A (60.5%), B (59.0%), and C (54.1%), respectively. The RDI of cyclophosphamide in arm B was 61.5%. A total of 2 (5%), 3 (8%), and 6 (16%) did not complete the priming phase due to disease progression during the first 12 weeks in arms A, B, and C, respectively.

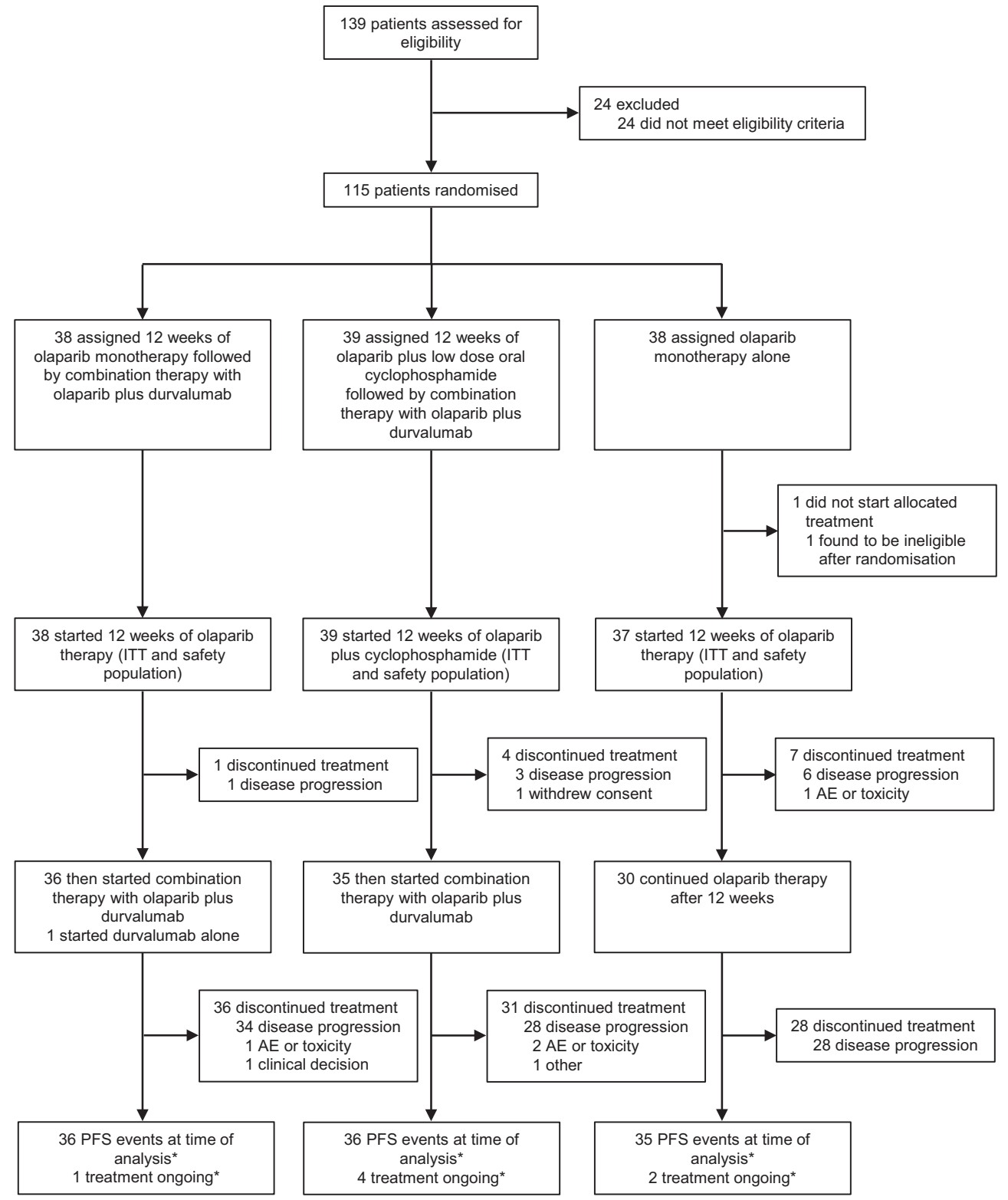

Fig. 1 | CONSORT Diagram. PFS progression free survival, AE adverse event. *Cut-off date July 10th, 2024.

## Efficacy

After a median follow-up time of 44.7 months (range, 1-47 months), progression events were observed in 94.7%, 92.3% and 94.6% in arms A, B, and C, respectively. The 36-week PFS rates were 47.4% (95% CI 31.0-62.1), 48.7% (95% 32.5-63.2), and 35.1% (95% CI 20.4-50.3), respectively. The median PFS were 35.6 weeks (95% CI 23.6-40.4), 35.9 weeks (95% CI 23.7-48.1), and 24.4 weeks (95% CI 22.1-36.1), respectively (Fig. 2A).

The confirmed objective response rate (ORR) according to Response Evaluation Criteria in Solid Tumors (RECIST) and the Gynecologic Cancer Intergroup (GCIG) CA-125 was 42.1% (95% CI 26.3%-59.2%) for arm A, 53.8% (95% CI 37.2%-69.9%) for arm B, and 35.1% (95% 20.2%-52.5%) for arm C. In the 102 participants (89.5%) with measurable disease, Fig. 3 illustrates the waterfall plots and the percentage of maximal change in the sum of lesion size from baseline.

**Table 1 | Baseline characteristics**

|  | Arm A N = 38 (%) | Arm B N = 39 (%) | Arm C N = 37 (%) |
|---|---|---|---|
| Median Age (range) | 65 (42-83) | 63 (44-81) | 72 (46-87) |
| ECOG PS 0 | 31 (81.6) | 32 (82.1) | 27 (73.0) |
| Ovarian primary | 32 (84.2) | 31 (79.5) | 22 (59.5) |
| FIGO Stage III at diagnosis | 22 (57.9) | 25 (64.1) | 27 (73.0) |
| FIGO Stage IV at diagnosis | 11 (29.0) | 11 (28.2) | 3 (8.1) |
| Measurable disease | 27 (71.1) | 28 (71.8) | 26 (70.3) |
| PFI > 12 months | 25 (65.8) | 24 (61.5) | 24 (64.9) |
| CA125 > 100 | 23 (60.5) | 24 (61.5) | 22 (59.5) |
| HRD *BRCA* wild-type | 19 (50.0) | 20 (51.3) | 19 (51.4) |
| HRD *BRCA* mutation | 3 (7.9) | 5 (12.8) | 5 (13.5) |
| HRP *BRCA* wild-type | 10 (26.3) | 10 (25.6) | 9 (24.3) |
| CUP-CC- | 15 (42.9) | 16 (44.4) | 15 (29.7) |
| CUP-CC+ | 20 (57.1) | 20 (55.6) | 18 (54.6) |

*ECOG PS* Eastern Cooperative Oncology Group Performance Status, *FIGO* Federation of Gynecology and Obstetrics, *PFI* Platinum-free Interval, *CA125* Cancer Antigen 125, *HRP* homologous recombinant proficient, *HRD* homologous recombinant deficient, *CUP-CC* CCR4 up-regulation, cytokines and chemokines.

## Treatment outcomes by homologous recombination deficiency status

A total of 100 (87.7%) out of 114 participants who had sufficient tumor purity underwent HRD testing, including tissue *BRCA1/2* mutation. There were 13% (N = 13) with *BRCA1/2* mutation, 29% (N = 29) with HRP, and 58% (N = 58) were HRD and *BRCA* wild-type. A proportion of the 106 cases tested found to have promoter methylation of either *BRCA1* or *RAD51C*. A high proportion of promoter methylation (defined as >70%) of *BRCA1* was present (N = 9, 8.5% of cases tested). No cases with *RAD51C* promoter methylation were found to have high level promoter methylation ( > 70%).

Amongst participants with HRD EOC (*BRCA* mutated, N = 13; *BRCA* wild-type, N = 58), the 36-week PFS rates were 59.1% (95% CI 36.1¬76.2), 56.0% (95% 34.8-72.3), and 37.5% (95% CI 19.0-56.0) for arms A, B, and C, respectively (Fig. 2B). Amongst participants with HRP EOC (*BRCA* wild-type, N = 29), the 36-week PFS rates were 40.0% (95% CI 12.3-67.0), 40.0% (95% 12.3-67.0), and 22.2% (95% CI 3.4-51.3), respectively (Fig. 2C).

For HRD EOC, confirmed ORR by RECIST only was 57.9% (95% CI 33.5%-79.7%), 63.6% (95% CI 40.7%-82.8%), and 36.4% (95% 17.2%-59.3%), respectively. For HRP EOC, confirmed ORR by RECIST only was 22.2% (95% CI 2.8%-60.0%), 40.0% (95% CI 12.2%-73.8%), and 14.3% (95% 0.4%-57.9%), respectively. Figure 3 provides the waterfall plots reporting the percentage of maximal change in sum of lesion size from baseline in 89 participants (78.1%) with measurable disease and known HRD status.

Across all treatment arms, there was significant difference in PFS between HRD and HRP subgroups (hazard ratio (HR) stratified by treatment arms, 0.55, 95% CI 0.35-0.87, p = 0.01; Fig. 4A).

## Correlative analysis

To have an agile tool to study the immunomodulatory in vivo changes following cyclophosphamide treatment in study participants, we developed an in vitro assay. In healthy blood donors (N = 24), stimulation with mafosphamide, an active metabolite of cyclophosphamide, of peripheral blood mononuclear cells (PBMC), for 72 h in vitro, increased the ratios of CD4 + CD25- T cell (Teff) to Treg (mean ratio pre- vs post- mafosphamide, calculated as change in the ratios between mafosphamide and untreated controls tested with Wilcoxon matched-pairs signed rank tests; 1.927 vs 1.000, p < 0.0001) and of CD8 T cell to

Treg (1.438 vs 1.000, p < 0.0096) (Supplementary material of the SOLACE2 trial: Supplementary note 1, Supplementary Fig. 2). These data indicated that this assay might be useful to predict whether an individual would have an immunological response following LDCy treatment. Upon culture of PBMC with mafosphamide in healthy donors, CCR4 expression was also significantly upregulated on CD4 + CD25- T cells (1.281 vs 1.000, p < 0.0001) and CD8 + T cells (1.492 vs 1.000, p = 0.0008), but not on Treg (0.997 vs 1.000, p = 0.8334), calculated as changes in the ratios between mafosphamide and untreated controls and tested with Wilcoxon matched-pairs signed rank tests. Migration, analysed by the CUP transwell migration assay, demonstrated a strong correlation between mafosphamide-induced CCR4 upregulated T cells and the migration index towards CCL22 (r = 0.81, p < 0.001). These findings indicate CCR4 upregulation by LDCy would similarly confer increased migration capability for specific T cell subsets.

In the SOLACE2 trial, in vitro testing of pre-treatment PBMC with mafosphamide, evaluable in 100 (87.7%) participants, also resulted in increases in the ratios of CD4 + CD25- T cell/Treg, (2.626 vs 1.000, p < 0.0001) and CD8 T cell/Treg (2.492 vs 1.000, p = 0.0002). When the participants were categorized based on the mean cut-point of 270 for post-mafosphamide CD4 + CD25- T cell/Treg ratio, there was no significant PFS difference between these groups (logrank p = 0.56; Supplementary material of the SOLACE2 trial: Supplementary note 1, Supplementary Fig. 3A). Similarly, there was also no significant PFS difference for the post-mafosphamide CD8 T cell/Treg ratio, based on the mean cut-point of 40 (logrank p = 0.68; Supplementary material of the SOLACE2 trial: Supplementary note 1, Supplementary Fig. 3B). These in vitro mafosphamide analyses were performed on PBMC which had been collected prior to commencement of trial therapy, and the analysis was performed on all evaluable cases regardless of treatment arm assignment.

Assessment of upregulation of CCR4 following in vitro mafosphamide stimulation of PBMCs, was evaluable in 106 (93.0%) SOLACE2 participants, the majority of whom upregulated CCR4 on both CD4 + CD25- T cells, (when normalized to untreated controls, 1.396 vs 1.000, p < 0.0001, Wilcoxon matched-pairs signed rank test) and CD8 + T cells (1.413 vs 1.000, p < 0.0001), but not on Tregs (1.088 vs 1.000, p = 0.0851) (Supplementary material of the SOLACE2 trial: Supplementary note 1, Supplementary Fig. 4). Participants who had a higher than mean percentage of CCR4+ effector T cells within the mafosphamide culture, had a 34% reduction in the risk of progression or death when stratified by treatment arms (stratified Hazard Ratio (HR 0.66, 95% CI 0.44-1.00, p = 0.05; Supplementary material of the SOLACE2 trial: Supplementary note 1, Supplementary Fig. 5). The median PFS for those participants with high CCR4 proportions in effector T cells (CUP + ) vs those below the mean (CUP-) were 43.0 weeks (95% CI 24.9-59.4) and 23.9 weeks (95% CI 23.4–36.1), respectively.

Baseline sera were evaluable in 105 (92.1%) SOLACE2 participants. As compared with healthy blood donors, SOLACE2 participants had significantly higher levels of IL-6 (mean 5.3 vs 3.9 pg/mL, p = 0.0399), IL-8 (mean 6.6 vs 3.8 pg/mL, p < 0.0001) and CCL17 pg/ml (mean 122.1 vs 54.11, p < 0.001), and lower CCL22 levels (mean 307.1 vs 412.2 pg/mL, p = 0.0131) (Supplementary material of the SOLACE2 trial: Supplementary note 1, Supplementary Fig. 6). When the SOLACE2 participants were categorized based on the mean distributions of these biomarkers, baseline high IL-6 ( > 5.3 pg/mL) was associated with inferior PFS (stratified HR 1.99, 95% CI 1.29-3.07, p = 0.002; Supplementary material of the SOLACE2 trial: Supplementary note 1, Supplementary Fig. 7A). Similar findings were observed for a high baseline IL-8 ( > 6.6 pg/mL) (stratified HR 1.41, 95% CI 0.93-2.14, p = 0.11; Supplementary material of the SOLACE2 trial: Supplementary note 1, Supplementary Fig. 7B). Baseline CCL17 (logrank p = 0.65; Supplementary material of the SOLACE2 trial: Supplementary note 1,

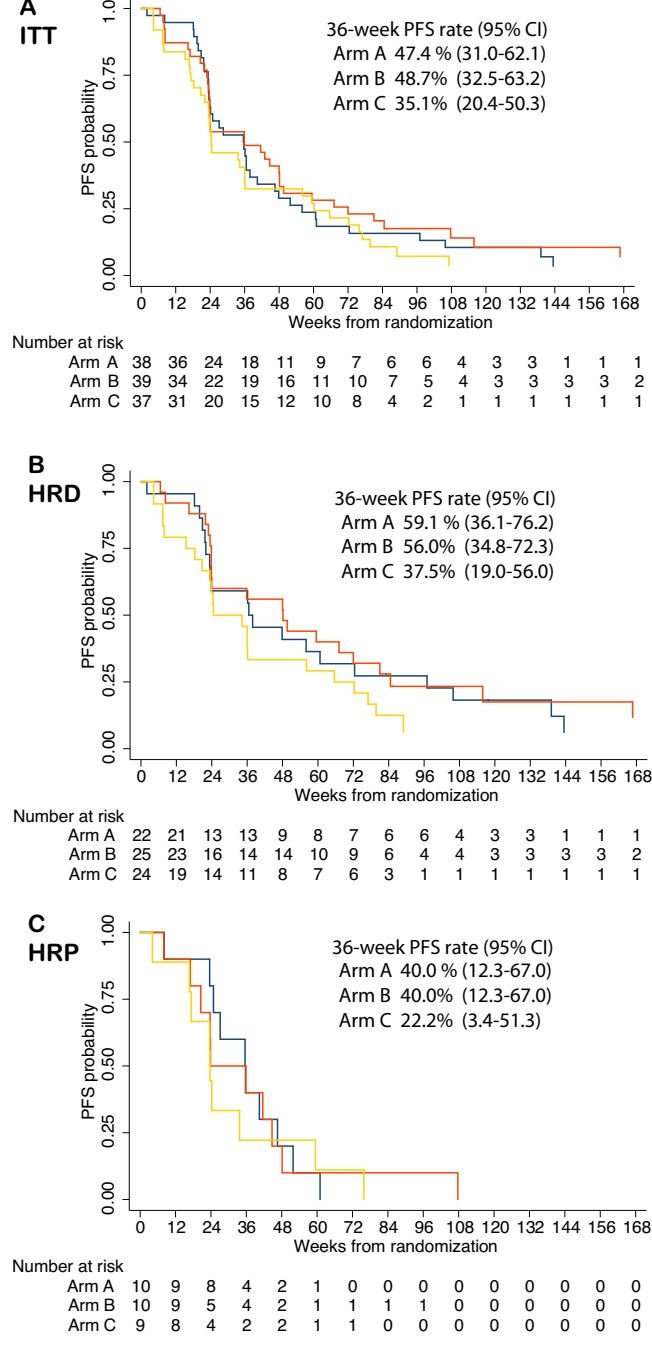

**Fig. 2 | Progression free survival in intention-to-treat population and key subgroups. A** Kaplan-Meier plot of progression-free survival by treatment arms in the intention-to-treat population. **B** Kaplan-Meier plot of progression-free survival by treatment arms in the homologous recombination-deficient population. **C** Kaplan-Meier plot of progression-free survival by treatment arms in the homologous recombination proficient population. Arm A olaparib followed by olaparib-durvalumab, Arm B olaparib-LDCy followed by olaparib-durvalumab, Arm C olaparib monotherapy, ITT Intention-to-treat- population, PFS progression-free survival, CI confidence interval, HRD homologous recombination deficient, HRP homologous recombination proficient. PFS was estimated using Kaplan-Meier methodology, and a two-sided 95% CI for 36-week PFS rate was calculated with the log-log transformation method. Source data are provided as a Source Data file.

Supplementary Fig. 7C) and CCL22 levels (logrank p = 0.83; Supplementary material of the SOLACE2 trial: Supplementary note 1, Supplementary Fig. 7D) were not significantly associated with PFS.

The CUP-CC assay was developed based on a combined algorithm that quantitatively combined CCR4 upregulation, CCL17 and CCL22 levels, as well as IL-6 and IL-8 levels, with the formula including CCR4 expressed as a percentage, cytokines in ng/mL, and chemokines as ng/mL. A total of 104 (91.2%) participants were evaluable for the combined CUP and IL-6 and IL-8 cytokines level (CUP-C). For CUP-C + , n = 54 and CUP-C-, n = 50. CUP-C+ was associated with significant reduction in the risk of progression or death (stratified HR 0.39, 95% CI 0.26-0.60, p < 0.001; Supplementary material of the SOLACE2 trial: Supplementary note 1, Supplementary Fig. 8). The median PFS rates were 48.0 weeks (95% CI 36.1- 65.6) and 23.4 weeks (95% CI 21.9-23.9) for CUP-C+ and CUP-C- respectively. Supplementary material of the SOLACE2 trial: Supplementary Note 1, Supplementary Fig. 8 demonstrates that the prognostic impact of CUP-C did not differ by treatment arm.

The same 104 (91.2%) participants were evaluable for CUP-CC inclusive of CUP, cytokines (IL-6 and IL-8) and chemokines (CCL17 and CCL22). For CUP-CC + , n = 58 and CUP-CC-, n = 46. In those for whom HRD status was also known (92/114, 80.7%), there were 11 (20.4%), 15 (27.8%) and 28 (51.9%) *BRCA* mutated, HRP, and HRD/*BRCA* wild-type EOC amongst CUP-CC+ participants, respectively. Amongst CUP-CC- participants, there were 2 (5.3%), 12 (31.6%) and 24 (63.2%), respectively. CUP-CC+ was associated with significant reduction in the risk of progression or death (stratified HR 0.31, 95% CI 0.19-0.49, p = 4.517 × $10^{-7}$; Fig. 4B). The median PFS rates were 49.6 weeks (95% CI 36.0-72.3) and 23.4 weeks (95% CI 21.9-24.1) for CUP-CC+ and CUP-CC- respectively for all treatment arms combined. The prognostic impact of CUP-CC did not differ by treatment arm (Fig. 5). Supplementary material of the SOLACE2 trial: Supplementary Note 1, Supplementary Fig. 9 demonstrates that the prognostic impact according to CUP-CC status was maintained with the exclusion of *BRCA* mutated participants.

In multivariable analysis, CUP-CC status remained a statistically significant variable for PFS (HR 0.37, 95% CI 0.22-0.61, p = 0.0001) after adjustment of known baseline prognostic factors in recurrent PSROC (Table 2).

We further evaluated the joint prognostic values of HRD and CUP-CC status, in predicting PFS. Significant differences were found in these groups (logrank p = 2.55 × $10^{-6}$; Fig. 4C) with median PFS for HRP cases, of 23.6 weeks (95% CI 8.1-46.6) for HRP/CUP-CC- (n = 12) versus 35.6 weeks (95% CI 23.7-44.7) for HRP/CUP-CC+ (n = 15), and for HRD cases, of 23.9 weeks (95% CI 21.9-34.3) for HRD/CUP-CC- (n = 26) versus 72.0 weeks (95% CI 36.1-89.0) for HRD/CUP-CC+ (n = 39).

Regardless of treatment arm, CUP-CC+ was also associated with a higher ORR (RECIST plus GCIG CA-125) than CUP-CC- (58.6% vs 28.3%, p = 0.002). There were no significant differences in ORR across treatment arms by CUP-CC status. Figure 3 provides the waterfall plots that report the percentage of maximal change in sum of lesion size from baseline in 95 participants (83.3%) with measurable disease and known CUP-CC status. Thirteen cases had a particularly long ( > 24 months) time on treatment (Fig. 6). Additional molecular information, including the cause of HRD, where it was identifiable, as well as evidence of high tumor mutational burden (TMB, >20 mutations per megabase) is provided in Supplementary material of the SOLACE2 trial: Supplementary note 1, Supplementary Fig. 10. Data for tumor TMB, categorized based on a cutoff of 5 mutations per megabase (Supplementary material of the SOLACE2 trial: Supplementary note 1, Supplementary Fig. 11), and plasma IFN-γ levels, categorized based on the mean distribution (Supplementary material of the SOLACE2 trial: Supplementary note 1, Supplementary Fig. 12) are also provided. Neither TMB nor IFN-γ levels were significantly associated with PFS.

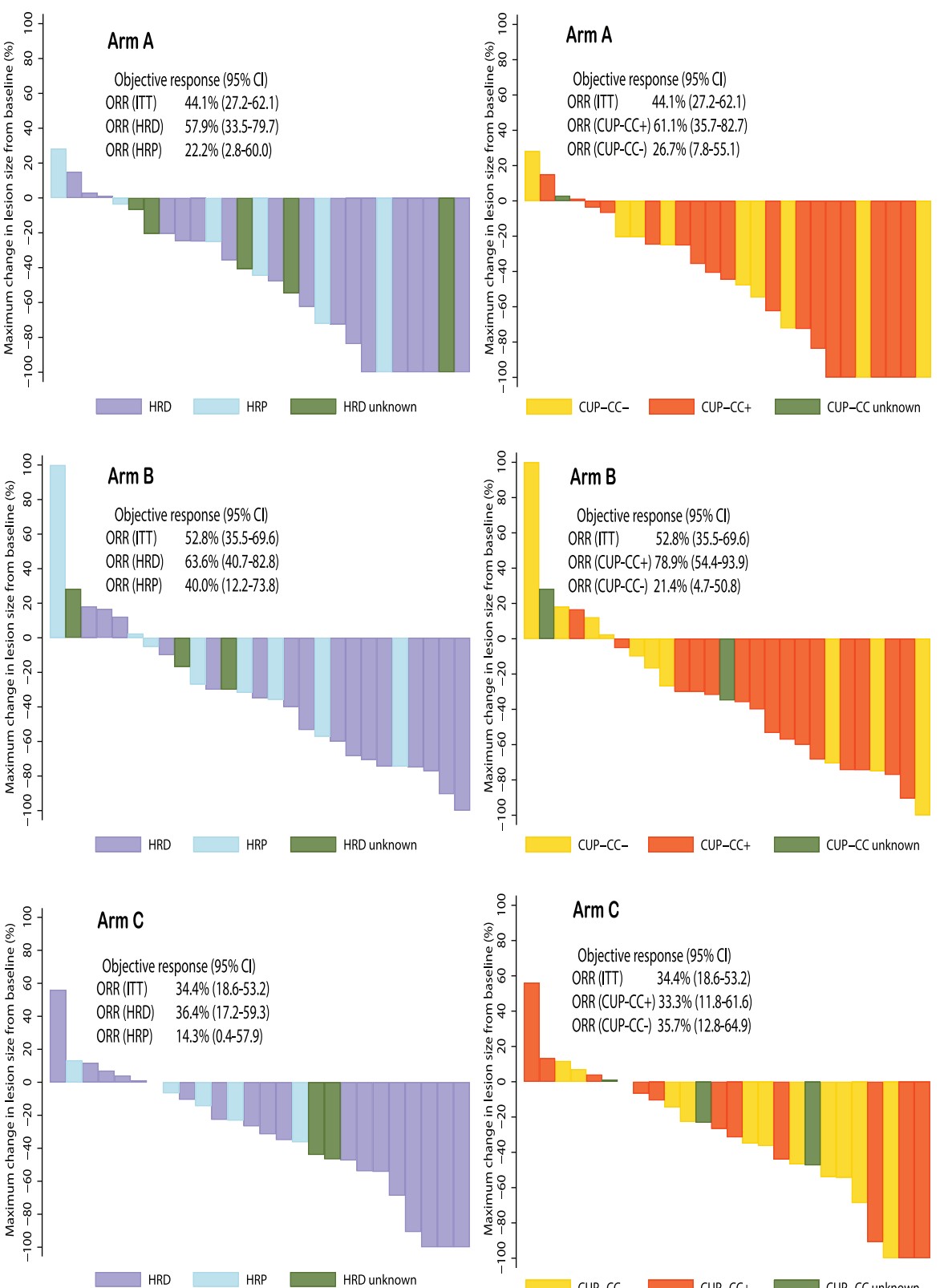

**Fig. 3 | Waterfall plots by treatment arms and biomarker subgroups.** Waterfall plots of maximum percentage change in sum of target lesions from baseline by treatment arms, homologous recombination deficiency status and CUP-CC status. Arm A olaparib followed by olaparib-durvalumab, Arm B olaparib-LDCy followed by olaparib-durvalumab, Arm C olaparib monotherapy, ITT Intention-to-treat population, HRD homologous recombination deficient, HRP homologous recombination proficient, ORR objective response rate, CI confidence interval, CUP-CC, CCR4 up-regulation, cytokines and chemokines ORR and corresponding 95% two-sided CI were calculated using the exact method.

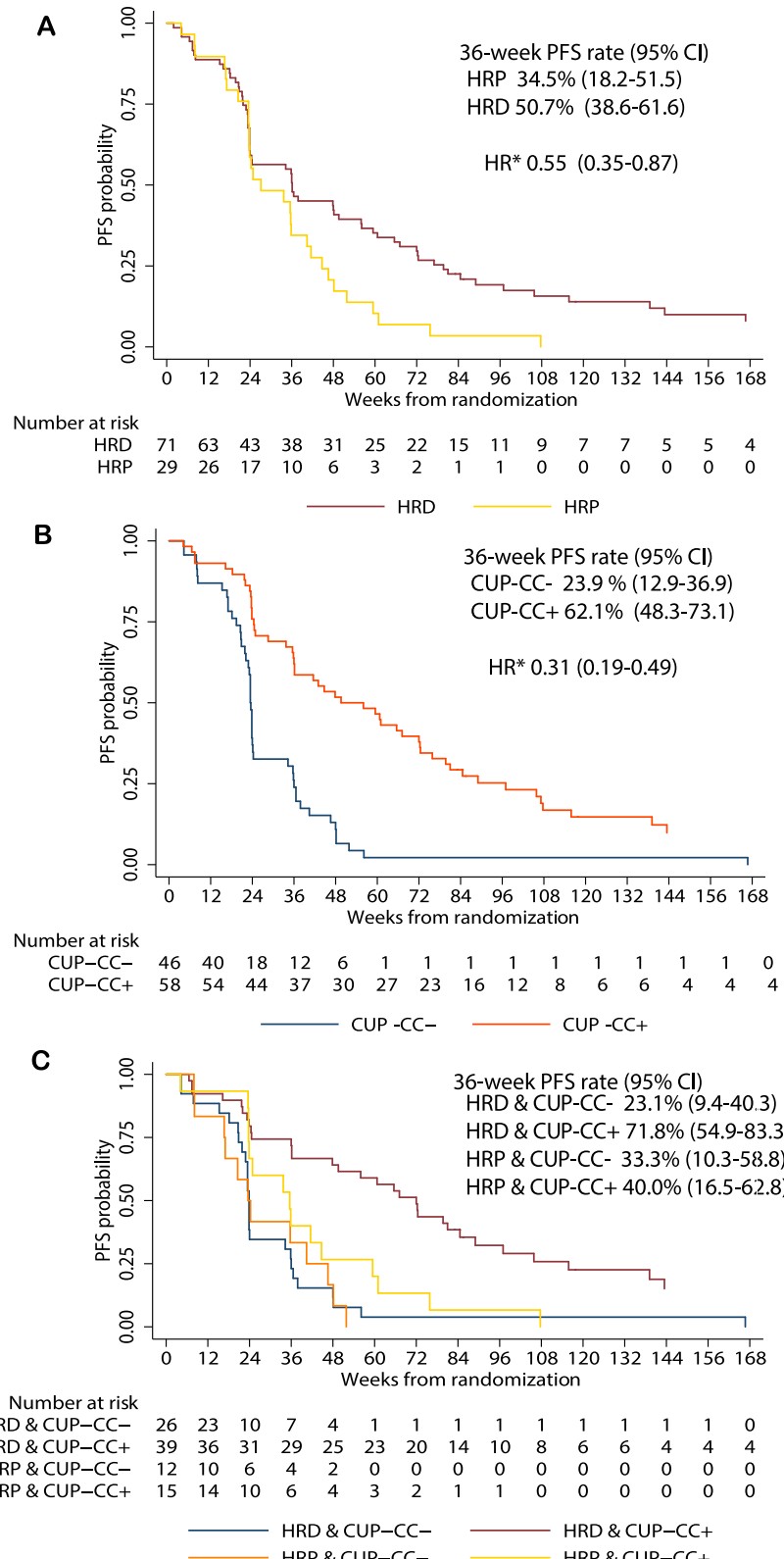

**Fig. 4 | Progression free survival in biomarker subgroups.** Kaplan-Meier plots of progression-free survival according to (**A**) homologous recombination deficiency status, (**B**) CUP-CC status, and (**C**) combined statuses of homologous recombination deficiency and CUP-CC Arm A, olaparib followed by olaparib-durvalumab; Arm B, olaparib-LDCy followed by olaparib-durvalumab; Arm C olaparib monotherapy, PFS progression-free survival, CI confidence interval, HRD homologous recombination deficient, HRP homologous recombination proficient, HR* hazard ratio, CCUP-CC, CCR4 up-regulation, cytokines, and chemokines. PFS was estimated using Kaplan-Meier methodology, and a two-sided 95% CI for 36-week PFS rate was calculated with the log-log transformation method. The HR and corresponding two-sided 95% CI were estimated using a Cox proportional hazards model, stratified by treatment arms. Source data are provided as a Source Data file.

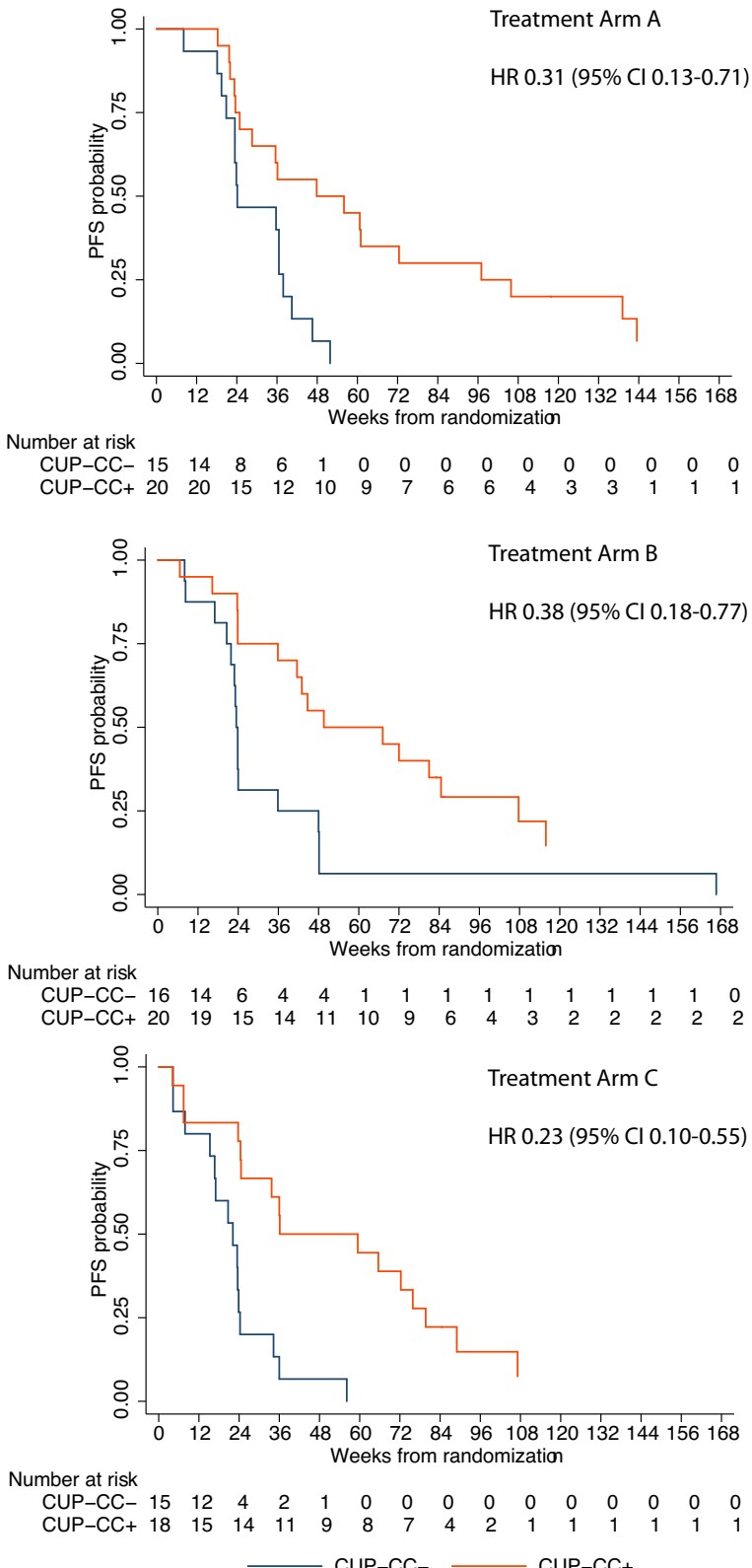

**Fig. 5 | Progression-free survival by treatment arms and CUP-CC status.** Kaplan-Meier plots of progression-free survival according to CUP-CC status in different treatment arms Arm A, olaparib followed by olaparib-durvalumab, Arm B olaparib-LDCy followed by olaparib-durvalumab, Arm C olaparib monotherapy, HR hazard radio, CI confidence interval, HRD homologous recombination deficient, HRP homologous recombination proficient, CUP-CC, CCR4 up-regulation, cytokines and chemokines. PFS was estimated using Kaplan-Meier methodology. The HR and corresponding two-sided 95% CI were estimated using a Cox proportional hazards model, stratified by treatment arms. Source data are provided as a Source Data file.

**Table 2 | Univariable and multivariable analyses to evaluate prognostic significance of CUP-CC**

| Variables | Univariable analysis[a] | | | | Multivariable analysis[a] (N = 92) | | | |
|---|---|---|---|---|---|---|---|---|
| | N | HR | 95% CI | | $P$[b] | HR | 95% CI | | $P$[b] |
| CUP-CC- | 46 | 1.00 | | | | 1.00 | | | |
| CUP-CC+ | 58 | 0.31 | 0.19 | 0.49 | $4.52 \times 10^{7}$ | 0.37 | 0.22 | 0.61 | 0.0001 |
| ECOG PS 0 | 90 | 1.00 | | | | 1.00 | | | |
| ECOG PS 1 | 24 | 0.52 | 0.31 | 0.86 | 0.011 | 0.59 | 0.31 | 1.10 | 0.095 |
| CA125 < 100 | 45 | 1.00 | | | | 1.00 | | | |
| CA125 ≥ 100 | 69 | 1.12 | 0.75 | 1.66 | 0.590 | 1.18 | 0.72 | 1.92 | 0.516 |
| PFI 6-12 moths | 41 | 1.00 | | | | 1.00 | | | |
| PFI > 12 months | 73 | 0.49 | 0.32 | 0.75 | 0.0013 | 0.43 | 0.24 | 0.77 | 0.005 |
| Non-measurable | 33 | 1.00 | | | | 1.00 | | | |
| Measurable | 81 | 1.03 | 0.67 | 1.57 | 0.899 | 1.67 | 0.93 | 2.99 | 0.083 |
| HRP | 29 | 1.00 | | | | 1.00 | | | |
| HRD | 71 | 0.55 | 0.35 | 0.87 | $2.55 \times 10^{6}$ | 0.62 | 0.38 | 1.02 | 0.061 |

[a]Cox proportional hazard regression analysis stratified by treatment arms.

[b]Statistical tests were two-sided. No adjustments were made for multiple comparisons.

*ECOG PS* Eastern Cooperative Oncology Group Performance Status, *PFI* Platinum-free Interval, *CA125* Cancer Antigen 125, *HRP* homologous recombinant proficient, *HRD* homologous recombinant deficient, *CUP-CC* CCR4 up-regulation, cytokines and chemokines.

## Safety

During the priming phase, grade 3 or higher adverse events (AEs) were reported in 15.8%, 43.6% and 21.6% in arms A, B, and C, respectively (Table 3). During the consolidation phase, grade 3 or higher AEs were reported in 37.8%, 48.6% and 40.0%, respectively. During both the priming and consolidation phases, the most commonly reported AEs of any grade were nausea, fatigue, and anemia (Supplementary data-sets of the SOLACE2 trial: Supplementary Table 1, Supplementary Table 2). Grade 3 or higher immune-related AEs (irAEs) were only reported in one participant (2.9%) from Arm B (Supplementary data-sets of the SOLACE2 trial: Supplementary Table 3). Any grade AE that led to permanent study treatment discontinuation occurred only in two participants (5.4%), during priming in arm C, and in another two participants (5.7%) in arm B during consolidation. One participant (arm B) developed myelodysplastic syndrome and transformed to acute myeloid leukemia. Another participant (arm B) developed pancreatic cancer (arm B, *BRCA2* mutation carrier).

Preplanned outcomes that are not reported here include the secondary objectives of health-related quality of life, time to start of first subsequent therapy, and time to the development of symptoms associated with progression, including abdominal/gastrointestinal symptoms. Tertiary objectives not yet reported include baseline PDL1 expression and the relationship to response to durvalumab and the value of ctDNA in predicting treatment benefit. These results are planned to be reported in a separate quality of life publication and a publication exploring and validating the translational outcomes of the trial.

## Discussion

The SOLACE2 trial was designed to determine whether PARPi-based priming of immune cells could improve the efficacy of subsequent olaparib-durvalumab in a PARPi naïve population. All participants received first-line platinum-based chemotherapy at the time of initial diagnosis and were then predominantly asymptomatic at the time of study enrollment. Priming was tested with either olaparib alone or with the combination of olaparib-LDCy based on the trial primary endpoint of 36-week PFS rate. The choice of this study endpoint was based on our hypothesis that 12 weeks of priming with either strategy, followed by 24 weeks of olaparib- durvalumab maintenance, would demonstrate an improvement in PFS at 36 weeks. However, neither intervention arms A (36-week PFS rate 47.4%) nor B (36-week PFS rate 48.7%) met the threshold of a 36-week PFS rate of 67%. This threshold

was based on a 20% improvement from an historical 36-week PFS rate of 47% observed with single-agent rucaparib[26]. Our olaparib mono-therapy arm had a worse outcome than expected, with a 36-week PFS rate of 35.1%. Baseline characteristics are consistent with prior studies. In the setting of our clinical selection of participants with no immediate indication for chemotherapy, we observed a predominance of participants with HRD cancers (71%).

At the time of study planning, there was evolving evidence to support the role of maintenance PARPi treatment in PSROC following response to chemotherapy, independent of *BRCA* status as platinum sensitivity was considered a good surrogate for likelihood of response to PARPi therapy[27–30]. We had also performed a meta-analysis to demonstrate that although there was greater PFS benefit with maintenance PARPi in the HRD over the HRP subpopulations, there was still significant PFS prolongation in the HRP subpopulation when treated with PARPi over placebo[31]. The role of PARPi therapy in the HRP population was not well-defined at the time and remains controversial, and furthermore, routine genetic testing was largely limited to germline *BRCA*. For these reasons, our study did not restrict enrollment of study participants by HRD/HRP status. There was also emerging evidence from the ARIEL2 study that the reported median PFS with single agent rucaparib was 5.7 months in the loss of heterozygosity (LOH) high subgroup which did not differ significantly from 5.2 months in the LOH low subgroup[26]. Subsequently, data on the activity of olaparib plus durvalumab was reported in the phase 2 MEDIOLA trial[32] that showed ORR of 34.4% and median PFS of 5.5 months in the cohort of *BRCA* wild-type patients treated with olaparib-durvalumab with PSROC, consistent with the findings from our HRP subpopulation. There is also similarity in the outcomes of the germline *BRCA* mutant cohort of MEDIOLA with our study findings. Both MEDIOLA and our study were recruiting participants at similar times and hence the performance of olaparib-durvalumab was unknown at our study planning. Finally, SOLACE2 was also co-evaluating the utility of CUP-CC test and hence the olaparib monotherapy arm was crucial to differentiate between prognostic versus predictive effect of this biomarker regarding the addition of ICI to PARPi.

In subgroup analysis, we did not observe statistically significant PFS differences by treatment arms according to HRD status. In the olaparib monotherapy arm, the 36-week PFS rate was 37.5% for the HRD population, which was numerically greater than the HRP population with a 36-week PFS rate of 22.2%. Within the HRD population, ORR by RECIST only for olaparib monotherapy was 36.4%. In contrast,

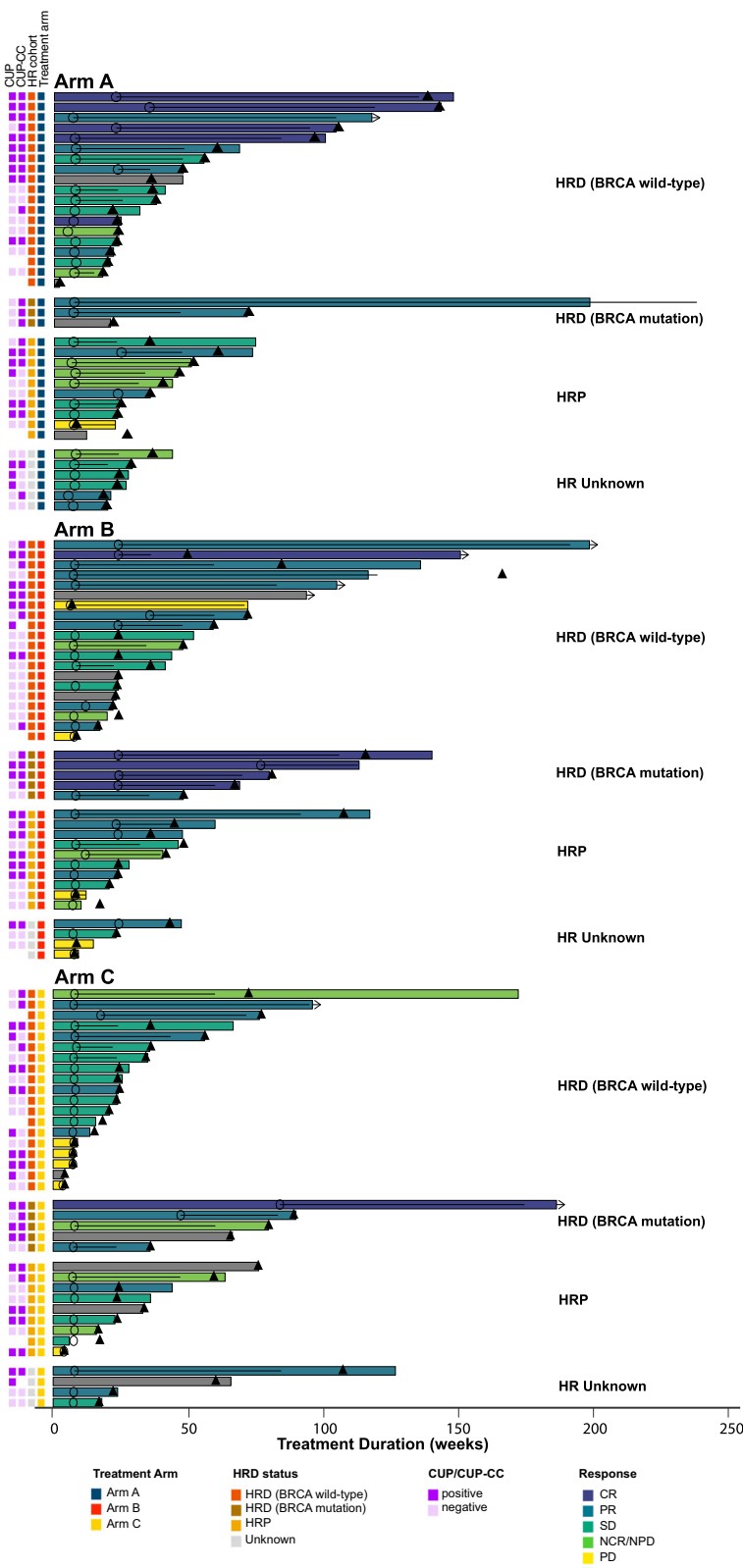

**Fig. 6 | Swimmer plot by treatment arms.** Swimmer plot of individual participant trajectories over time CUP, CCR4 up-regulation; CUP-CC, CCR4 up-regulation, cytokines and chemokines; HR, Homologous recombination; Arm A, olaparib followed by olaparib-durvalumab; Arm B olaparib-LDCy followed by olaparib-durvalumab, Arm C olaparib monotherapy, HRD homologous recombination deficient, HRP homologous recombination proficient, BRCA breast cancer gene, CR complete response, PR partial response, SD stable disease, NCR/NPR Non-CR/Non-PD, PD Progressive disease.

**Table 3 | Summary of adverse events (AE)**

| | ARM A (N = 38) | ARM B (N = 39) | ARM (N = 37) |
|---|---|---|---|
| Priming phase | | | |
| AE of any grade | 36 (94.7%) | 39 (100.0%) | 34 (91.9%) |
| Grade 3 + AE | 6 (15.8%) | 17 (43.6%) | 8 (21.6%) |
| AE leading to dose interruption - Any grade | 2 (5.3%) | 5 (12.8%) | 2 (5.4%) |
| AE leading to dose reduction - Any grade | 0 | 0 | 0 |
| AE leading to permanent discontinuation - Any grade | 0 (0.0%) | 0 (0.0%) | 2 (5.4%) |
| | ARM A (N = 37) | ARM B (N = 35) | ARM C (N = 30) |
| Consolidation phase | | | |
| AE of any grade | 36 (97.3%) | 35 (100.0%) | 29 (96.7%) |
| Grade 3 + AE | 14 (37.8%) | 17 (48.6%) | 12 (40.0%) |
| AE leading to dose interruption - Any grade | 2 (5.4%) | 3 (8.6%) | 4 (13.3%) |
| AE leading to dose reduction - Any grade | 0 | 0 | 0 |
| AE leading to permanent discontinuation - Any grade | 0 (0.0%) | 2 (5.7%) | 0 (0.0%) |

ORR for olaparib-LDCy-durvalumab and olaparib-durvalumab were 63.6% and 57.9%, respectively. As expected, participants with *BRCA1/2* mutation had prolonged PFS regardless of treatment arm.

A key pre-defined correlative objective was to evaluate the utility of the CUP-CC assay in determining the outcome of PARPi-based treatments. This assay reflects the up-regulation of the chemokine receptor, CCR4, on T cells after in vitro treatment with mafosphamide (the active metabolite of LDCy), coupled with baseline levels of the cognate CCR4 chemokine ligands, CCL17 and CCL22, and circulating cytokines IL-6 and IL-8. The CUP-CC test demonstrated robust prognostic ability for patients undergoing olaparib-based therapy. Across all treatment arms, the median PFS was 49.6 weeks for CUP-CC+ and, in contrast, only 23.4 weeks for CUP-CC-; ORR was higher in CUP-CC+ than in CUP-CC- (58.6% vs 28.3%, p = 0.002). In 13 participants with sustained benefit, receiving more than 24 months of study treatments, 12 were classified as CUP-CC + . The CUP-CC test was not predictive of additional PFS benefit of LDCy (arm B vs arm A), or durvalumab (arms A or B vs arm C).

These consistent findings across all treatment arms, and in both HRD and HRP subpopulations, is unexpected since the CUP assay was designed to evaluate immune reactivity to an analogue of cyclophosphamide, not a PARPi. This finding should nevertheless be considered in the context of the CUP assay uncovering an individual's overall T cell (beyond Tregs) response to diverse biological signals, resulting in upregulation of CCR4, and promotion of T cell migration into the TME. Higher ratios of effector T cells/Treg in the CUP assay in patients with cancer compared to controls suggest cancer-associated differences in baseline reactivity, supporting the biological utility of the CUP assay in patients. A number of these signals, including histone modifications, can be impacted by PARPi therapy[33,34].

Importantly, the utility of the CUP-CC assay may be to characterize the current HRD status of the cancer. In our trial, tissue HRD status of 87.7% participants was based on archival FFPE tissue with adequate tumor purity. Genomic scar tests, such as Myriad CDx, classify the cancer based on both current and prior DNA damage that occurred during tumor evolution, but does not inform about acquired resistance, which could result in a functional change from an HRD to an HRP status, a key factor in resistance to PARPi therapy. For example, in ARIEL2[26], differences in the results of LOH assays were reported between pre-treatment biopsies versus archival tumor materials,

demonstrating that although 34% of patients with LOH-low cancers based on archival tissue biopsy were reclassified as LOH-high based on the pre-treatment biopsy, no cases were observed to reverse from LOH-high status to LOH-low status. The conclusion of the ARIEL2 translational studies was that accumulation of genomic scarring is an irreversible process, persisting even as cancers re-acquire functional HRR[35]. The CUP-CC blood-based assay has the potential to address these challenges.

Validation of these findings is currently ongoing to confirm the utility of the CUP-CC assay in other clinical trial datasets. Since SOLACE2 participants were all treated with olaparib-based therapies, the predictive value of this assay for olaparib benefit could not be demonstrated, and hence, work is ongoing in the maintenance setting of PARPi versus placebo, as well as evaluation for different combination therapies, including PARPi-ICI and PARPi-bevacizumab. The utility of this assay will also be compared against approved HRD tests.

We have previously demonstrated that *RAD51C* and *RAD51D* mutations and high-level *BRCA1* promoter methylation predict for PARPi response[35,36]. In this study, some high-level *BRCA1* promoter methylated participants had poor responses to olaparib which may be due to loss of methylation in one or more of the *BRCA1* alleles, or acquired a distinct mechanism of PARPi resistance. Therefore, our ongoing work to evaluate the change of CUP-CC status throughout treatment and its ability to predict either the loss of HRD or other causes for PARPi resistance is crucial to further personalize therapy.

Limitations of the SOLACE2 trial include its modest sample size which restricts the statistical power to detect meaningful differences between treatment arms. Furthermore, tumor response assessments were conducted without blinding, introducing the potential for bias in the evaluation of efficacy outcomes. However, correlative analyses involving immune biomarkers and tumor genetics were performed with blinding to both treatment allocation and clinical outcomes. While the prognostic potential of the CUP-CC assay is compelling, further validation is necessary to establish its clinical utility.

In conclusion, the SOLACE2 trial showed that neither immune priming strategy was definitively superior to olaparib monotherapy, as the PFS differences were not statistically significant and failed to meet our pre-specified historical threshold of efficacy. Nevertheless, olaparib-durvalumab and olaparib-LDCy-durvalumab were associated with numerically greater ORR and longer PFS as compared with olaparib monotherapy but this study was not powered for relative comparison between treatment arms. Most importantly, we characterized concise immunological features of study participants who benefited the most from these olaparib-based therapies, including in the HRP and BRCA wild-type subgroup. Ongoing work will continue to better identify patients and treatment strategies to optimally incorporate PARPi and ICI into the treatment paradigm for PSROC. Further quality of life and translational analyses will be undertaken and are planned for publication in future manuscripts.

## Methods
### Ethics statement
SOLACE2 was a multicentre non-comparative phase 2 trial conducted at 15 Metropolitan and Regional Hospitals across Australia (ACTRN12618000686202). SOLACE2 was conducted in accordance with the Good Clinical Practice guidelines of the International Council for Harmonization and the Declaration of Helsinki. Independent ethics committees approved participation information and consent forms at each site. All participants provided signed informed consent. Ethical approval was provided by Sydney Local Health District Human Research Ethics Committee - RPAH Zone.

### Participants
Eligible participants with histologically confirmed high-grade serous carcinoma of the ovary, fallopian tube or primary peritoneum,

underwent prior surgery and received one line of platinum-based chemotherapy. Participants must have had PSROC, with asymptomatic or minimally symptomatic disease, raised CA-125 (≥70 kU/L) and/or measurable disease.

Other key eligibility criteria included Eastern Cooperative Oncology Group (ECOG) performance status of 0 or 1 and adequate end organ function. Participants were excluded if they had prior use of a PARPi, active autoimmune or inflammatory disorders, prior diagnosis of myelodysplastic syndrome or acute myeloid leukemia. The full eligibility criteria are available in the study protocol (Supplementary material of the SOLACE2 trial: Supplementary note 2). Participant recruitment commenced in February 2019. No participants were recruited prior to trial registration.

## Study treatment

Participants were randomly assigned 1:1:1 to three treatment arms, stratified by site/institution, platinum-free interval (6-12 months vs >12 months), CA-125 (<100 U/mL vs ≥100 U/mL), measurable disease (present vs absent), and germline *BRCA* mutation status (present vs absent).

During the priming phase (first three cycles following random assignment, with a cycle interval of every 28 days), participants in arm B received LDCy orally 50 mg for days 1–5 every week. All participants in arms A, B, and C received olaparib 300 mg twice daily continuously. During the consolidation phase (cycle 4 or day 85 onwards), participants in arms A and B received durvalumab 1500 mg (fixed dose) once every 28 days for up to 3 years. All participants in arms A, B, and C continued to receive olaparib 300mg twice daily continuously. Treatment continued until disease progression, unmanageable toxicity, or study withdrawal. Treatment beyond radiological progression with olaparib and durvalumab were allowed.

Before the database lock on 10th July 2024, tumor testing of tissue samples was performed using an inhouse proprietary HRD test. An immune based signature was also developed using a CUP test and four circulating chemokines and cytokines (CUP-CC).

This investigator-initiated study was led by the Australia and New Zealand Gynecological Oncology Group, the Walter and Eliza Hall Institute of Medical Research, the RMIT University, and the National Health and Medical Research Council Clinical Trials Centre, University of Sydney.

## Endpoints and assessments

The primary endpoint was PFS rate at 36 weeks. PFS was defined as the time from random assignment to objective disease progression (RECIST v1.1) or death. Tumor assessments were performed using computed tomography at baseline, at the end of weeks 8 and 24, then every 12 weeks until disease progression.

Key secondary endpoints included objective response rate by RECIST and the GCIG CA-125, patient-reported outcome, and time to commencement of subsequent therapy. Adverse events (AEs) were graded according to the National Cancer Institute Common Terminology Criteria for Adverse Events (version 5.0). Key tertiary endpoints included the prognostic and predictive potential of exploratory biomarkers in predicting treatment response.

## Statistics & reproducibility

To compute the sample size, we used the Fleming one-stage design with a null hypothesis of a 36-week PFS rate of 47% and an alternative hypothesis of 67% for the primary endpoint with 80% power and 5% one-sided type I error. For each of the combination treatment arms A and B, 38 participants per arm will be required. Treatment arm C served as a concurrent non-comparative control. If both combination treatment arms are active, the play-the-winner selection strategy by Simon, Wittes, and Ellenberg[37] will be applied to select the superior arm where a 15% minimum difference from the best performing

combination regimen will be able to be detected with a probability >80%. The best performing regimen will then be selected to inform a larger comparative study. The trial would recruit a total of 114 participants.

Participants were randomly assigned to one of the three treatment arms. Investigators and participants were not blinded to treatment arm allocation. All participants who received at least one cycle of therapy and had their disease re-evaluated beyond baseline were considered evaluable for response. All participants who received at least one dose of therapy were considered evaluable for safety.

PFS was estimated using the Kaplan-Meier method according to the intention-to-treat principle. The HR and corresponding two-sided 95% CI were estimated using a Cox proportional hazards model, stratified by treatment arms. A two-sided 95% CI for the 36-week PFS rate was calculated with the log-log transformation method. ORRs were calculated using the exact method. Safety results are presented for each treatment arm. Prognostic and predictive of *BRCA* mutation, HRD, HRP, CUP, TMB, cytokines, chemokines, and CUP-CC were evaluated. All exploratory analyses were considered descriptive with no adjustment for multiplicity. Statistical analysis was undertaken using STATA version 15.1. The Swimmer's plot in Fig. 6 was developed using R version 4.2.2.

## Whole exome sequencing and homologous recombination deficiency analysis

Archival formalin-fixed paraffin-embedded (FFPE) tumor samples (either chemotherapy naïve or following neo-adjuvant chemotherapy) were retrieved for all participants. An H&E-stained section was reviewed by a pathologist, and regions of highest tumor purity were marked for extractions. Tumor DNA (and RNA) were isolated using the Omega Bio-tek Mag-Bind® FFPE DNA/RNA Kit (M6955-00); and germline DNA was extracted from whole blood using the QIAamp DNA mini kit (Qiagen; 51304). Library preparation for whole exome sequencing (WES) was performed with Twist Exome v2 (Twist Biosciences, USA) with IDT xGen cfDNA & FFPE library prep (Integrated DNA Technologies, Coralville, USA) with Unique Molecule Identifiers and using mechanical shearing. Libraries were sequenced to a target depth of 20Gbp with 150PE reads (200x) on the Illumina Novaseq X plus platform. Library preparation and sequencing was performed by the Australian Genome Research Facility (Melbourne, AUS).

Tumor and matched germline DNA were analysed using a BioNix pipeline (https://github.com/PapenfussLab/bionix)[38] as follows. Reads with duplicate UMIs were removed using dedumi (Bedo, manuscript in preparation https://github.com/PapenfussLab/dedumi) and aligned to GRCh38p31[39] with StrobeAlign (0.11.0)[40]. Octopus (v0.7.0)[41] was used to call and phase SNVs and indels with subsequent annotation against Ensembl (v110)[42] within regions ±100 bp of exon boundaries. Variants were annotated with dbNSFP (v4.2a)[43,44] using SnpEff and SnpSift (v4.3t)[45]. Copy number variants were called using FACETS (0.6.1)[45]. HRD status was called following Marquard et al.[46]. using their published code updated to GRCh38 and applied to copy number calls from FACETS.

Tumor mutational burden (TMB) was estimated as the number of passing non-synonymous somatic variants with a haplotype frequency >20% divided by 36 Mbp to normalize for the length of the analysed region.

## Variant curation

Germline variant lists collated from processed WES data were ordered by variant allele frequency (AF) plus the presence of the variant in the Familial Cancer Centre (FCC) database (i.e. highest AF with an FCC entry first). Variants in the blood with allele frequencies lower than 50% (i.e., below the expected AF for heterozygosity) were not investigated further. Loss of heterozygosity of a germline variant in the tumor was determined by a change in AF in the tumor sample compared to the blood.

Collated somatic variants were ordered by haplotype frequency (HF) plus the presence of the variant in the COSMIC database (i.e. highest HF with a COSMIC entry first). Variants were investigated from the highest HF (usually a percentage equivalent to the purity of tumor) down to an HF of ~10%.

The likelihood of pathogenicity for each variant of interest was determined using information from the following databases: OncoKB (https://www.oncokb.org/); Clinvar (https://www.ncbi.nlm.nih.gov/clinvar/) and the GENIE Cohort v15.1-public database (https://genie.cbioportal.org/). Evidence for the pathogenicity of copy number variants (CNVs) was determined using information from OncoKB. Only transcript variants or CNVs with clear evidence of pathogenicity were reported for each case. Those with conflicting evidence of pathogenicity were classed as variants of potential significance.

## CCR4 up-regulation, chemokines and cytokines (CUP-CC) assay

Cryopreserved peripheral blood mononuclear cells (PBMCs) from trial participants and healthy adult female donors during routine blood donations, acquired at the Australian Red Cross Blood Service were thawed and cultured with in complete AIM V media alone or with mafosphamide (Niomech, Germany), at 4.5 µg/ml for 72 h at 37 °C in a 5% $CO_2$ humidified incubator. Following incubation, cells were washed and surface cell CCR4 expression on diverse T cell subsets determined using flow cytometry staining with a cocktail of antibodies including anti-CD3-AF700 (Biolegend, Cat.No. 300424, clone UCHT1, dilution 1:100), anti-CD8-PerCP-Cy5.5 (Biolegend, Cat.No. 560662, clone RPA-T8, dilution 1:100), anti-CD4-APC-Cy7 (Biolegend, Cat.No. 300518, clone RPA-T4, dilution 1:100), anti-CD25-PE-Dazzle594 (Biolegend, Cat.No. 356126, clone M-A251, dilution 1:50), anti-CCR4 BV421 (Biolegend, Cat.No. 359414, clone L291H4, B381005, 1:50). Additional biomarkers were added to further validate subsets or gating including anti-FoxP3-APC (Invitrogen, Cat.No. 17-4776-42, clone PCH101, dilution 1:40) and anti-CD127-BV650 (Biolegend, Cat.No. 351326, clone A019D5, dilution 1:100) for Treg (not shown). A fixable live/dead cell dye (zombie-aqua cell viability kit, Biolegend, Cat. No. 423102, dilution 1:500) distinguished dead vs live cells. The intracellular FoxP3 was stained following fixation and permeabilization of cells using a fixation/permeabilization buffer kit (eBioscience, USA). Flow cytometry data were acquired on a ZE-5 cell analyzer (Bio-Rad, USA) using the Everest software (collecting a minimum of 200,000 events per sample). Fluorescence minus one (FMO) controls were used to enable accurate gating. The percentage of CD3 + CD4 + CD25- T cell effector cells as well as CD3 + CD8 + T cells expressing CCR4 after mafosphamide culture (flow cytometry gating shown in Supplementary Fig. 2) was determined using Flowjo software (TreeStar, USA). Percentages were added to determine the CUP value; the mean of all CUP values was used as the threshold to determine CUP+ (above mean) vs CUP- (under mean).

The cytokines and chemokines were measured from plasma using the Bioplex Multiplex Luminex assays (Bio-Rad) following the manufacturer's protocol for the premix beads of IL-6 (171-BK29MR2), IL-8 (171-BK31MR2), CCL22 (171-BK41MR2), CCL17 (171-BK53MR2), and IFNγ (171-BK25MR2). The data were acquired and analysed using the BioPlex-200 (Bio-Rad). The mean values of the cytokines and chemokines levels from plasma samples of SOLACE2 participants were used to categorize these participants as "high" (above mean) vs "low" (below mean).

The CUP value derived above was combined with CCL22, CCL17, IL6, and IL8 (chemokines and cytokines; CC), with individual values adjusted to the same scale according to the formula CUP-CC=CUP + ((CCL22+CCL17)/10 − (IL6+IL8)*5).

## Reporting summary

Further information on research design is available in the Nature Portfolio Reporting Summary linked to this article.

## Data availability

Deidentified study data are available for sharing. To request access to the deidentified study data, please contact the Corresponding Author. Confirmation of receiving the request will be made within two weeks. Requests will be reviewed by the Trial Management Committee and written applications from investigators with the academic capability and credibility to undertake the work proposed will be considered. The scientific merit of the proposal, including the appropriate methods, analysis, and publication plan will be assessed. Consideration will be taken of any overlap with analyses already undertaken or planned to be undertaken by the study team. If a proposal is approved, a signed data transfer agreement will be required before data sharing. The duration of access to the data will be according to the data transfer agreement. Source data are provided with this paper.

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

## Acknowledgements

We thank the participants, their families and carers, research staff at participating hospitals, the NHMRC Clinical Trials Centre of The University of Sydney, and the Australia New Zealand Gynecological Oncology Group. We thank Ratana Lim, Rachael Taylor and Jocelyn Penington for technical assistance. Momodou Cox, Nirashaa Bound, Hina Amer and Emily Cassar were supported by PhD Scholarships awarded by RMIT University. Emily Cassar was also supported by an Australian Government Research Training Program Scholarship. This work was supported by the NHMRC Clinical Trials Centre Program Grant, the Australian Cancer Research Foundation, the Victorian State Government Operational Infrastructure Support, Australian Government NHMRC IRIISS, fellowships and grants from the Cancer Council Victoria (Sir Edward Dunlop Fellowship in Cancer Research (CLS), Stafford Fox Medical Research Foundation (C.L.S. and M.J.W.), the American Association of Cancer Research (AACR-AstraZeneca Ovarian Cancer Research Fellowship 2022 (K.N.)), Rivkin Center for Ovarian Cancer 2020 Rosser Family Pilot Study Award (M.J.W.), MRFF 2021 Genomic Health Futures Mission (M.J.W.), NHMRC (MP and AK), ANZGOG-OASIS (AK, CD), strategic funding via the RMIT University Enabling Impact Platform (A.K.,C.De., G.G., M.P.), and philanthropic anonymous donations (M.A.Q., M.M., M.P.). The University of Sydney is the sponsor of this study, with the role of initiating and managing this trial, ensuring

regulatory compliance, protecting participants' safety, ensuring data integrity and quality assurance, and overall oversight of this trial. This trial also received funding from AstraZeneca and its Group of Companies. The study design, data collection and analysis, and manuscript writing was done independently of AstraZeneca.

## Author contributions

Conception and design: C.K.L., M.F., C.L.S., and M.P. Development of methodology: C.K.L., M.F., C.L.S., M.P., A.K., N.B., M.M., M.A.Q., M.J.W., C.J.V., K.S.A and J.B. Acquisition of clinical data (acquired and managed patients, provided facilities, etc.): C.K.L., M.F., C.L.S., Y.C.L., J.L., S.B.H., Y.A., C.S., S.S.N., and P.B. Analysis and interpretation of data (statistical analysis, biostatistics, computational analysis): C.K.L., R.L.O.C., M.P., A.K., A.T.P., M.J.W., C.J.V., and K.E.F., K.S.A., J.B. and C.L.S. Administrative, technical, or material support (i.e., reporting or organizing data, constructing database): K.E.F., K.D., C.D.a, C.D.e, G.G., D.Z., Y.C.L., S.S., A.L., K.S.A., J.B., K.N., M.C., A.J., E.C., H.A., U.G.K. Writing, review, and/or revision of the manuscript: All authors. Study supervision: C.K.L., M.F., C.L.S., and M.P.

## Competing interests

CKL reports grants from Roche, Amgen, AstraZeneca and Merck KGaA, honoraria from AstraZeneca, Amgen, Janssen, GSK, Boehringer Ingelheim, MSD, Roche, Gilead, Novartis and Glenmark and meeting attendance support from AstraZeneca and Janssen. KEF reports honoraria from Janssen and Gilead, and meeting attendance support from BMS, Merck/MSD, Gilead, and Pfizer. KN reports support for the present manuscript with an AACR-AstraZeneca Ovarian Cancer Research Fellowship, and support from the Stafford Fox Medical Research Foundation. MM reports a US patent, and AstraZeneca stock as an employee incentive. YCL reports institutional grants form BeiGene, honoraria from AstraZeneca and Eisai, and participation in an advisory board with AstraZeneca and GSK. JL reports honoraria from AstraZeneca, Eisai, Gilead, Novartis and GSK, meeting attendance support from GSK and Novartis, participation in an advisory board with AstraZeneca, and a leadership role as chair of the ANZGOG uterine tumor working group. SB-H reports honoraria from AstraZeneca, Gilead, GSK, Novartis, MSD and Eisai, meeting attendance support from Novartis, and reports a leadership role as a member of the ANZGOG ovarian cancer tumor working group. YA reports consulting fees from Eisai, GSK, AstraZeneca, Pfizer, MSD and Daiichi Sankyo, honoraria from GSK, AstraZeneca, MSD, Eisai and IQVIA, meeting attendance support from MSD, participation in and advisory board with GSK, Eisai, AstraZeneca, MSD and Pfizer, and a leadership role through participation on the ANZGOG Research Advisory Committee. CS reports honoraria from AstraZeneca, Gilead, GSK, Eli Lilly, Novartis, MSD and Eisai, participation in and advisory board with AstraZeneca, and a leadership role as a member of ANZGOG ovarian cancer working group. SS-N reports honoraria from AstraZeneca, MSD and Eisai, meeting attendance support from MSD and a leadership role on the South Australian Health Cancer Drug Committee. PB reported participation in an advisory board with GSK and AstraZeneca. DZ reports honoraria from Merck and Bayer. MAQ reports participation in an advisory board with GSK. ATP reports support for the current manuscript through and NHMRC investigator grant and a leadership role as past president of the Australian Bioinformatics and Computational Biology Society Inc. CV reports support for the current manuscript with support from the Stafford Fox Medical Research Foundation and from Victorian Medical Research Acceleration Fund, institutional grants from AstraZeneca, Boehringer Ingelheim, IDEAYA Biosciences and Eisai and the recipient of royalties from Venetoclax. MF reports investigator-initiated grant support for the study from AstraZeneca, consulting fees from AstraZeneca, Novartis, GSK, Incyclix, AbbVie and BioNTech, honoraria from AstraZeneca, GSK, MSD and Limbic, participation in an advisory board from AGITG IDSMB, Endo-3 trial and Domenica trial, and institutional grants from AstraZeneca, BeiGene and Novartis. CLS reports support for the current manuscript through an investigator led grant from AstraZeneca, an NHMRC investigator grant, support from the Stafford Fox Research Foundation, the OASIS philanthropic research grant from ANZGOG, and the Victorian Medical Research Acceleration Fund, research grants from AstraZeneca, Eisai, BeiGene, Sierra Oncology, Boehringer Ingelheim and IDEAYA, royalties from Navitoclax, Genentech, Abbott, and Walter and Eliza Hall Institute of Medical Research, meeting attendance support from AstraZeneca, advisory board participation with AstraZeneca, Eisai, Sierra Oncology, MSD and Takeda, a leadership role as GCIC director, chair of the board of the international rare cancer initiative, and chair of the board of ANZGOG, and drug provision as part of a research agreement from AstraZeneca, Clovis Oncology, Eisai, BeiGene, Sierra Oncology, Boehringer Ingelheim and Ideaya.MP reports support for the current manuscript through AstraZeneca clinical trial funding support, RMIT and WEHI translational research funding, and ANZGOG translational research funding, and a patent provisionally lodged on blood biomarkers to identify PARPi responders. All other authors (A.J., A.L., A.E.R.K., N.T.B., C.Da, C.De, E.C., G.G., H.A., U.G.K., J.B., K.D., K.S.A., M.C., M.J.W., R.L.O.C., and S.S.) declare no competing interests.

## Additional information

Chee Khoon Lee [1,2] ✉, Apriliana E. R. Kartikasari[3,20], Nirashaa T. Bound [3,4,20], Katherine E. Francis [1,5], Kristy Shield-Artin [4,6], Justin Bedo [4,6], Ksenija Nesic [4,6], Katrina Diamante[1], Rachel L. O'Connell[1], Mutsa Madondo[3], Momodou Cox[3], Claire Davies [7], Cyril Deceneux[3], Georgia Goodchild[3], Andrew Jarratt[4], Emily Cassar [3], Hina Amer [3], U. G. Imalki U. Kariyawasam [4,6], Yeh Chen Lee[1,5,8], Janine Lombard[9], Sally Baron-Hay [10], Yoland Antill[11,12], Catherine Shannon[13], Sudarshan Selva-Nayagam[14], Philip Beale[15], Danka Zebic[1], Sandy Simon [1], Anneliese Linaker[1], Michael A. Quinn[16], Anthony T. Papenfuss [4,6], Matthew J. Wakefield [4,6,17], Cassandra J. Vandenberg [4,6], Michael Friedlander [5,8,20], Clare L. Scott [4,6,17,18,19,20] & Magdalena Plebanski [3,20]

[1]NHMRC Clinical Trials Centre, The University of Sydney, Sydney, NSW, Australia. [2]Department of Medical Oncology, St George Hospital, Kogarah, NSW, Australia. [3]Accelerator for Translational Research and Clinical Trials (ATRACT) Centre, School of Health and Biomedical Sciences, RMIT University, Bundoora, VIC, Australia. [4]Walter and Eliza Hall Institute of Medical Research, Parkville, VIC, Australia. [5]Department of Medical Oncology, Prince of Wales Hospital and Royal Hospital for Women, Randwick, NSW, Australia. [6]Department of Medical Biology, University of Melbourne, Parkville, VIC, Australia. [7]Australia and New Zealand Gynaecological Oncology Group, Camperdown, NSW, Australia. [8]School of Clinical Medicine, Faculty of Medicine and Health, UNSW, Sydney, NSW, Australia. [9]Newcastle Private Hospital, New Lambton Heights, NSW, Australia. [10]Department of Medical Oncology, Royal North Shore Hospital, St Leonards, NSW, Australia. [11]Department of Medical Oncology, Peninsula Health, Frankston, VIC, Australia. [12]Faculty of Medical, Dental and Health Sciences, Monash University, Clayton, VIC, Australia. [13]Mater Cancer Care Centre, South Brisbane, QLD, Australia. [14]Department of Medical Oncology, Royal Adelaide Hospital, Adelaide, SA, Australia. [15]Department of Medical Oncology, Chris O'Brien Lifehouse, Camperdown, NSW, Australia. [16]Oncology Unit, Royal Women's Hospital and University of Melbourne, Melbourne, VIC, Australia. [17]Department of Obstetrics, Gynaecology and Newborn Health, University of Melbourne, Parkville, VIC, Australia. [18]Department of Medical Oncology, Peter MacCallum Cancer Centre and University of Melbourne, Parkville, VIC, Australia. [19]Royal Women's Hospital, Parkville, VIC, Australia. [20]These authors contributed equally: Apriliana E. R. Kartikasari, Nirashaa T. Bound, Michael Friedlander, Clare L. Scott, Magdalena Plebanski. ✉e-mail: chee.lee@sydney.edu.au

