## [Transparent Peer Review file · Nature Communications]

Olaparib, durvalumab, and cyclophosphamide, and a prognostic blood signature in platinum-sensitive ovarian cancer: the randomized phase 2 SOLACE2 trial

Corresponding Author: Professor Chee Lee

Version 0:

Reviewer comments:

Reviewer #1

(Remarks to the Author)

Key Noteworthy Results

The authors have described their non-comparative phase 2 trial investigating three different treatments for platinum sensitive recurrent ovarian cancer in PARP inhibitor naïve patients; 12 weeks of 'priming' treatment with Olaparib followed by Durvalumab-Olaparib, 12 weeks of 'priming' treatment with Olaparib-Cyclophosphamide followed by Durvalumab-Olaparib and Olaparib monotherapy. They found that both treatment arms containing durvalumab had prolonged PFS compared to single agent Olaparib, although the trial did not meet the pre specified PFS36 threshold.

The trial also investigated the use of a biomarker CUP-CC, a novel immune assay designed to evaluate reactivity to cyclophosphamide priming. The test was not predictive of response to cyclophosphamide priming, however, patients who were CUP-CC+ had a significantly longer PFS compared to those who were CUP-CC -. This difference was observed across all treatment arms irrespective of HRD status. This finding is of interest and further exploration/validation of this test as a biomarker for response to treatment with PARPi is warranted.

The study is novel but does not change standard of care and the results are not highly significant/relevant to the field

Methodology:

One big flaw in the study design is that lack of a Durvalumab - olaparib only arm with no priming as this would have provided a useful comparator to the experimental arms to answer the question whether the 12 weeks priming does increase response rates to olaparib and durvalumab. The study cannot answer this question which is the primary endpoint.

The authors based the stats/methodology around the historical response to single agent Rucaparib from the ARIEL 2 study. Hoping to see a 20% improvement in PFS36weeks. However a better comparison would have been the olaparib-durvalumab arm in the phase 2 MEDIOLA trial of similar patient population.

The use of PFS36weeks is also interesting as not a commonly used parameter for clinical response and the authors do not explain or justify its use as a primary endpoint.

It would be useful to discuss the rationale for choosing single agent Olaparib as a treatment arm in this study, particularly given the high proportion of HRP patients included. While the long terms survival outcomes from NOVA and ARIEL3 would not have been available at the time of study design and recruitment, their findings are relevant to SOLACE2. Both these studies call in to question the utility of monotherapy PARPi's for patients with BRCA1/2 wt PSROC. More information on why the olaparib only arm was included is needed in any revised manuscript.

Also the rationale for olaparib priming is not well explained and could be included in a revised manuscript.

Significance

The standard of care for PSROC remains platinum containing doublet containing chemotherapy with bevacizumab. The

OCEANS trial demonstrated a median PFS of 12.4m with Carboplatin-Gemcitabine-Bevacizumab in this setting. There is no mention of this in the introduction. This is also a chemotherapy sparing treatment and not maintenance and I think the authors need to make this clearer. Perhaps a figure in the main manuscript with the study design would be helpful. While SOLACE2 was a non-comparative trial, the median PFS demonstrated in all three treatment arms was less than 36 weeks (9 months). This is despite selecting patients that had 'no indication for immediate chemotherapy' and may have therefore been expected to have a more favorable prognosis.

The findings from this trial suggest that treatment with Durvalumab and Olaparib should not replace standard of care, regardless of treatment priming with Olaparib or Cyclophosphamide. As expected, single agent Olaparib was associated with a poor PFS in this study. Surprisingly, there was no significant difference in PFS between HRD and HRP subgroups across all treatment arms.

The finding of improved PFS for patients who were CUP-CC+ is interesting and warrants further investigation. As all patients in the trial received a Olaparib, it is not yet clear if this test will help to guide patient selection for PARPi – the test certainly appears to have promising prognostic value, but its predictive role remains uncertain.

Clarity and Context

The abstract could be clearer on several points. The first sentence of the abstract (lines 76-79) only describes two of the three treatment arms (the Olaparib only arm is not included) despite listing the number of patients as 114 (this number includes the 37 patients who received Olaparib alone). Lines 81-83 go on to state that priming with Olaparib or Olaparib + Cyclophosphamide was associated with a 'longer' PFS. It is unclear what the PFS is 'longer' in comparison to. Do the authors mean 'longer' compared to the single agent PARPi treatment arm (which has still not been mentioned at this point in the abstract)? Or 'longer' than historical PFS results from trials of durvalumab alone? Lastly, the authors have concluded in the abstract that CUP-CC may have the potential to personalize PARPi therapies for BRCA wt patients. As all patients in the trial received a PARPi it is too early to comment if this test will help to guide patient selection, although this will be an area of interest for future study.

References

As mentioned, a more robust discussion of the current treatment landscape for PSROC would be useful, with particular reference to the uncertain role of PARPi treatment for patients with BRCA1/2 wt status. Reference to other chemotherapy sparing upfront treatments in PROC setting is also needed as discussed.

Reviewer #2

(Remarks to the Author)

Reviewer #3

(Remarks to the Author)

In the SOLACE2 trial (ACTRN12618000686202), 115 patients with platinum-sensitive recurrent ovarian cancer were randomized 1:1:1 to receive olaparib followed by olaparib plus durvalumab (Arm A), olaparib plus low-dose cyclophosphamide followed by olaparib plus durvalumab (Arm B), or olaparib as continuous monotherapy (Arm C). 114 patients were evaluable for analysis. The primary endpoint was 36-week progression-free survival, with a null rate of 47% and target rate of 67%. Both intervention arms had a 36-week progression-free survival rate close to the null rate (Arm A: 47.4%, Arm B: 48.7%), with the authors noting that the noncomparative control arm also had a lower-than-expected 36-week progression-free survival rate (Arm C: 35.1%).

A key correlative objective was to evaluate the prognostic ability of the CUP-CC assay (CCR4 upregulation [CUP] combined with cytokines [IL-6 and IL-8] and chemokines [CCL17 and CCL22]) for patients receiving PARP inhibitor-based treatment, and whether it predicted additional progression-free survival benefit with durvalumab. CUP-CC status could not be determined in 10/114 (8.8%) patients; among the remaining patients, progression-free survival was significantly longer in CUP-CC+ patients than CUP-CC- patients (HR=0.31, 95% CI [0.19-0.49]), with median progression-free survival of 49.6 weeks for CUP-CC+ patients and 23.4 weeks for CUP-CC- patients. The objective response rate was 58.6% in CUP-CC+ patients and 28.3% in CUP-CC- patients (p=0.002). The CUP-CC assay was not predictive of additional progression-free survival benefit with durvalumab (i.e., Arm A or B versus Arm C). I have the following minor comments.

"In conclusion, the SOLACE2 trial shows olaparib-durvalumab and olaparib-LDCy-durvalumab are associated with numerically greater ORR and longer PFS as compared with olaparib monotherapy." (Lines 424-426). This conclusion seems inconsistent with the study design. Specifically, SOLACE2 was a noncomparative phase II trial (Lines 76 and 436), yet this conclusion focuses on between-arm comparisons (i.e., Arm A versus Arm C, Arm B versus Arm C). As originally designed, for the primary analysis, the Arm A and Arm B 36-week progression free survival rate would each be compared to a null rate of 47%—not to the rate observed in Arm C (which was described as a concurrent noncomparative control arm, Line 498).

Please clarify that this conclusion was not based on the intended primary analysis. Please also clarify that the objective response rate and progression-free survival were numerically but not statistically better in Arms A and B versus Arm C.

“The 36-week PFS rates were 47.4% (95% CI 31.0-62.1), 48.7% (95% 32.5-63.2), and 35.1% (95% CI 20.4-50.3), respectively” (Lines 196-197). The trial’s power calculation is based on a one-sided nominal significance level of 0.05, which would be consistent with 90% confidence intervals rather than 95% confidence intervals for these 36-week progression-free survival rates. Please update the confidence intervals or clarify.

The arm labels change in the statistical analysis section. Elsewhere in the manuscript Arm C is the control arm (i.e., olaparib as continuous monotherapy) whereas Arm A is the control arm in the statistical analysis section (Lines 496-498). Please update for consistency.

Lines 176 to 179 state that 114 patients were randomly assigned to Arm A, B, or C whereas the CONSORT diagram shows that 115 patients were randomly assigned. Please clarify in the manuscript text that 115 patients were randomly assigned to Arm A, B, or C but 1 patient was excluded post-randomization due to ineligibility (per the CONSORT diagram).

Reviewer #4

(Remarks to the Author)

General Feedback:

The SOLACE2 trial presents a comprehensive evaluation of PARPi-based immune priming in recurrent ovarian cancer, integrating translational biomarkers like CUP-CC to refine patient stratification. The study is methodologically rigorous, with well-defined endpoints and a strong emphasis on correlative analyses to explore immune modulation. The following issues can be addressed to improve the impact of this manuscript.

Specific Feedback:

Introduction:

1. The introduction lacks key epidemiological statistics for EOC and PSROC. Including incidence, mortality rates, survival data, and response rates to subsequent chemotherapy lines would provide a clearer clinical context.
2. Bevacizumab and PARP inhibitors are presented as treatment advances, but their limitations are not addressed. Discussing resistance mechanisms and predictive biomarkers would strengthen the rationale.
3. The section on CD3+/CD8+ TILs and Tregs is fragmented and not well connected to the study rationale. Clearly explain how Tregs suppress CD8+ TIL function and why overcoming this is key for ICI efficacy. Also, condense this discussion.
4. The introduction mentions that PARPi activates the STING pathway but does not convincingly explain why combining olaparib with durvalumab is expected to work in ovarian cancer. Acknowledge contradictory evidence, such as the JAVELIN Ovarian 200 trial failure, and explain how LDCy might enhance response.
5. If CUP-CC is a predictive biomarker, compare it to PD-L1 expression, tumor mutational burden, and IFN- γ signatures. Explain its potential advantage over these established markers.
6. Rather than stating that TILs improve survival, explain why their function is impaired in EOC (e.g., high Treg/CD8 ratio, immunosuppressive cytokines, restricted antigen presentation).

Results:

7. Discontinuation rates in the priming phase are reported, but baseline characteristics of these patients are not analyzed. Identifying if subgroups (e.g., HRD vs. HRP, CUP-CC+ vs. CUP-CC-) had higher discontinuation rates could offer insights into resistance.
8. CUP-CC status could not be determined in 8.8% of participants, but the reason is unclear. Clarify whether this was due to sample issues, patient refusal, or technical failures.
9. Waterfall plots show RECIST-defined responses, but cases of pseudoprogression or long-term responders not meeting RECIST criteria are not addressed. This is key for ICI-containing regimens, where pseudoprogression might be misclassified as progression.
10. The report notes that HRP cases with CUP-CC+ had significantly improved PFS compared to CUP-CC-, but it does not compare their outcomes to HRD patients. Clarify if CUP-CC+ can predict response independently of HRD.
11. Correlative analysis reports mafosfamide-induced T-cell changes in healthy donors but does not compare findings to patient samples. Such a comparison would validate its clinical relevance.
12. Only four patients discontinued due to AEs, which seems low given the reported toxicity. Clarify if dose reductions or treatment delays were needed to manage AEs.

Discussion:

13. The discussion states that neither intervention arm met the 36-week PFS goal but does not explore biological reasons for the lower-than-expected efficacy.
14. HRD status did not stratify PFS, but potential reasons (e.g., acquired resistance, tumor evolution) are not discussed.
15. ORR differences between treatment arms are noted but lack interpretation. The 60% ORR for olaparib/LDCy/durvalumab is much higher than the 38.5% for olaparib/durvalumab—does this suggest immune priming or just expected variability?
16. CUP-CC was not predictive of additional benefit from LDCy or durvalumab, but the discussion does not speculate on why immune priming failed to improve efficacy.
17. The claim that CUP-CC could expand PARPi use in HRP patients is interesting but lacks comparative data with existing biomarkers like PD-L1 expression and TMB. This would clarify its clinical utility.
18. HRD and HRP subgroups showed no significant PFS difference, yet the discussion does not explore why. Could high rates of HRD reversion mutations or epigenetic changes restoring homologous recombination proficiency be involved?
19. CUP-CC is described as a possible replacement for HRD genomic tests, but no direct evidence supports this. Can CUP-CC track treatment-induced changes in the tumor microenvironment over time?

Methods:

20. Was prior PARP inhibitor (PARPi) exposure an exclusion criterion, as prior exposure could influence response to Olaparib?

21. Why CA-125 (<100 vs. ≥100 U/mL) was used as a stratification factor over standard markers like treatment-free interval or platinum resistance scores?

22. Was the trial used blinded assessment for tumor response evaluations, as blinding is crucial to reduce bias in non-comparative trials?

Version 1:

Reviewer comments:

Reviewer #1

(Remarks to the Author)

The authors have provided satisfactory responses to my comments as detailed in the response letter and the manuscript has been revised accordingly

No additional comments

Reviewer #2

(Remarks to the Author)

Reviewer #3

(Remarks to the Author)

I would like to thank the authors for their responses to the comments in my review. I am satisfied with the authors' responses to my comments.

Reviewer #4

(Remarks to the Author)

AGREE WITH ALL CHANGES MADE. RECOMMEND APPROVAL

We thank the reviewers for their time and effort in critically reviewing our manuscript and for their comments and constructive feedback. We welcome the opportunity to respond to all the points raised and have revised our manuscript accordingly. We have italicized the comments from the reviewers below and provided our responses following each comment.

Reviewer comments:

Reviewer 1:

Reviewer #1 (ovarian cancer, clinical trial):

Key Noteworthy Results

The authors have described their non-comparative phase 2 trial investigating three different treatments for platinum sensitive recurrent ovarian cancer in PARP inhibitor naïve patients; 12 weeks of 'priming' treatment with Olaparib followed by Durvalumab-Olaparib, 12 weeks of 'priming' treatment with Olaparib-Cyclophosphamide followed by Durvalumab-Olaparib and Olaparib monotherapy. They found that both treatment arms containing durvalumab had prolonged PFS compared to single agent Olaparib, although the trial did not meet the pre specified PFS36 threshold.

The trial also investigated the use of a biomarker CUP-CC, a novel immune assay designed to evaluate reactivity to cyclophosphamide priming. The test was not predictive of response to cyclophosphamide priming, however, patients who were CUP-CC+ had a significantly longer PFS compared to those who were CUP-CC-. This difference was observed across all treatment arms irrespective of HRD status. This finding is of interest and further exploration/validation of this test as a biomarker for response to treatment with PARPi is warranted.

The study is novel but does not change standard of care and the results are not highly significant/relevant to the field

Authors' response: We thank the reviewer for pointing out that our study is novel and completely agree with the reviewer that the results of a relatively small, randomized phase 2 study, regardless of the trial findings, will not change standard of care and was never expected to be a practice changing trial. The purpose of our study was to test the multiple hypotheses which are all described in detail in our manuscript, with the aim to generate data to guide further research and potentially inform the design of a larger randomised trial depending on the research findings.

Methodology:

One big flaw in the study design is that lack of a Durvalumab - olaparib only arm with no priming as this would have provided a useful comparator to the experimental arms to answer the question whether the 12 weeks priming does increase response rates to olaparib and durvalumab. The study cannot answer this question which is the primary endpoint.

Authors' response: We carefully considered a durvalumab-olaparib only arm but decided against it because this combination was being investigated as part of the MEDIOLA trial which was also recruiting participants at the same time as our study. We specifically chose a control arm of olaparib monotherapy because we considered that it was the best comparator arm for this study, without any confounding effect of durvalumab on the immune system. With respect, we believe that this is a strength and not a flaw of our study design. One of the aims of SOLACE2 was co-evaluating the utility of the CUP-CC test and hence the olaparib monotherapy arm was crucial to differentiate between prognostic versus predictive effect of this putative biomarker, in relation to the addition of immunotherapy. All three arms contain olaparib and hence the predictive effect of the biomarker for olaparib is as yet unexplored. However, we have shown that this biomarker was prognostic for olaparib-based treatment.

The authors based the stats/methodology around the historical response to single agent Rucaparib from the ARIEL 2 study. Hoping to see a 20% improvement in PFS36weeks. However a better comparison would have been the olaparib-durvalumab arm in the phase 2 MEDIOLA trial of similar patient population.

Authors' response: We were unable to benchmark against MEDIOLA because these trial data were not available at the time of our study inception. Our study recruited participants between February 2019 to October 2022. The MEDIOLA trial recruited participants between May 4, 2018 and March 10, 2020.

The use of PFS36weeks is also interesting as not a commonly used parameter for clinical response and the authors do not explain or justify its use as a primary endpoint.

Authors' response: In our revised manuscript, we have provided the justification of our primary study endpoint as outlined below (page 15, paragraph 2):

“...Priming was tested with either olaparib alone or with the combination of olaparib-LDCy based on the trial primary endpoint of 36-week PFS rate. The choice of this study endpoint was based on our hypothesis that 12 weeks of priming with either strategy, followed by 24 weeks of olaparib-durvalumab maintenance, would demonstrate an improvement in PFS at 36 weeks. However, neither intervention arms A (36-week PFS rate 47.4%) nor B (36-week PFS rate 48.7%) met the threshold of a 36-week PFS rate of 67%. This threshold was based on a 20% improvement from a

historical 36-week PFS rate of 47% observed with single agent rucaparib²⁶. Our olaparib monotherapy arm had a worse outcome than expected, with a 36-week PFS rate of 35.1%. Despite baseline characteristics being consistent with prior studies, and our clinical selection of participants with no immediate indication for chemotherapy, in this smaller study, we observed a predominance of participants with HRP cancers (58%).....”

It would be useful to discuss the rationale for choosing single agent Olaparib as a treatment arm in this study, particularly given the high proportion of HRP patients included. While the long terms survival outcomes from NOVA and ARIEL3 would not have been available at the time of study design and recruitment, their findings are relevant to SOLACE2. Both these studies call in to question the utility of monotherapy PARPi's for patients with BRCA1/2 wt PSROC. More information on why the olaparib only arm was included is needed in any revised manuscript.

Authors' response: We thank the reviewer for the above comment. We have revised our manuscript to explain our rationale for including an olaparib monotherapy arm and to contextualise our manuscript to the time of study conception. In our revised manuscript, we have provided these explanations (page 16, paragraph 2):

“At the time of study planning, there was evolving evidence to support the role of maintenance PARPi treatment in PSROC following response to chemotherapy, independent of *BRCA* status as platinum sensitivity was considered a good surrogate for likelihood of response to PARPi maintenance therapy^{27, 28, 29, 30}. We had also performed a meta-analysis to demonstrate that although there was greater PFS benefit with maintenance PARPi in the HRD over the HRP subpopulations, there was still significant PFS prolongation in the HRP subpopulation when treated with PARPi over placebo³¹. The role of PARPi therapy in the HRP population was not well-defined at the time and remains controversial, and furthermore routine genetic testing was largely limited to germline *BRCA*. For these reasons our study did not restrict enrolment of study participants by HRD/HRP status. There was also emerging evidence from the ARIEL2 study that the reported median PFS with single agent rucaparib was 5.7 months in the loss of heterozygosity (LOH) high subgroup which did not differ significantly from 5.2 months in the LOH low subgroup²⁶. Subsequently, data on the activity of olaparib plus durvalumab was reported in the phase 2 MEDIOLA trial³² that showed ORR of 34.4% and median PFS of 5.5 months in the cohort of *BRCA* wild-type patients treated with olaparib plus durvalumab with PSROC, consistent with the findings from our HRP subpopulation. There is also similarity in the outcomes of the germline *BRCA* mutant cohort of MEDIOLA with our study findings. Both MEDIOLA and our study were recruiting participants at similar times and hence the performance of olaparib plus durvalumab was unknown at our study planning. Finally, SOLACE2 was also co-evaluating the utility of CUP-CC test and hence the olaparib monotherapy arm was crucial to differentiate between prognostic versus predictive effect of this biomarker regarding the addition of ICI to PARPi.”

Also the rationale for olaparib priming is not well explained and could be included in a revised manuscript.

Authors' response: Thank you for indicating that additional explanation regarding priming by olaparib is required. For clarity, we have included below, text that describes priming by PARPi, and also priming by LDCy, in the order in which it appears in the text.

We have included this on page 6, paragraph 2:

“.....Low dose cyclophosphamide treatment (LDCy) leads to Treg depletion and increased CD8 T cell/Treg ratios in the TME¹¹. We have previously conducted a phase 1 study and showed that the combination of olaparib and LDCy was tolerable and had promising clinical activity, particularly in *BRCA* mutated EOC¹². We therefore hypothesized that LDCy would help remodel the TME, priming the TME for optimal responsiveness to PARPi combination therapies.”

And on page 7, paragraph 1:

And on page 7, paragraph 1:

“.....Of relevance here, PARPi modulate the immune system through the stimulation of interferon genes (STING) pathway. PARPi induce DNA damage activates the cGAS-STING pathway, increasing IFN- γ release and enhancing T-cell-dendritic cell crosstalk and promoting antigen presentation^{14, 15}. PARPi have synergistic effects with ICI, through DNA damage, cell death, and neoantigen and other antigen release, thereby enhancing ICI and intrinsic immune responses, including enhanced antigen presentation, increasing tumor-infiltrating lymphocytes, upregulation of PD-L1, and reprogramming of other molecules and immune cells involved in the TME^{16, 17}. In this context, it is plausible that prior PARPi alone may prime the TME for subsequent PARPi-ICI therapy. In addition, an increased CD8 T-cell to Treg ratio promoted by LDCy¹⁸, could free the anti-tumor activity of the CD8 T-cells activated by PARPi treatment from Treg immunosuppression.”

And on page 7, paragraph 2:

“Data presented here demonstrate that LDCy further modulates the expression of CCR4 on conventional CD4+ and CD8+ T cells. On this basis, we further hypothesized that LDCy would prime the ovarian TME, through enrichment with effector CD4+ and CD8+ T cells, with their function subsequently enhanced by ICI treatment, and we developed an assay to study CCR4 up-regulation (CUP).....”

Significance

The standard of care for PSROC remains platinum containing doublet containing chemotherapy with bevacizumab. The OCEANS trial demonstrated a median PFS of 12.4m with Carboplatin-Gemcitabine-Bevacizumab in this setting. There is no mention of this in the introduction. This is also a chemotherapy sparing treatment and not maintenance and I think the authors needs to make this clearer. Perhaps a figure in the main manuscript with the study design would be helpful.

Authors' response: We thank the reviewer for raising the OCEANS trial data. We have included this in the introduction (page 5, paragraph 2):

“Platinum-sensitive recurrent high grade serous ovarian cancer (PSROC) has conventionally been defined as cancer progression ≥ 6 months after the most recent platinum-based chemotherapy. There is a 50% likelihood of response with retreatment with platinum-based chemotherapy, but there is a declining likelihood of chemotherapy response with each successive line of treatment³. The addition of bevacizumab increases the response rate to 70% and prolongs progression-free survival (PFS)⁴. However, the median overall survival (OS) is still less than 5 years, and similar with chemotherapy alone or addition of bevacizumab.”

As recommended, we have also created a new supplementary figure to display the study schema. We have also emphasized that our approach is chemotherapy-sparing as outlined below (page 8, paragraph 2):

“The SOLACE2 trial investigated two immune priming strategies, with either olaparib plus LDCy or olaparib monotherapy, followed by consolidation treatment with olaparib plus durvalumab to improve PFS in PSROC (Supplementary Figure 1). The chemotherapy-sparing interventions were offered as active therapies at the time of progression, rather than as maintenance treatment, for participants where chemotherapy was not yet indicated.....”

While SOLACE2 was a non-comparative trial, the median PFS demonstrated in all three treatment arms was less than 36 weeks (9 months). This is despite selecting patients that had ‘no indication for immediate chemotherapy’ and may have therefore been expected to have a more favorable prognosis.

Authors’ response: As indicated in our submitted manuscript, the baseline characteristics of our study population was consistent with other studies. As reflected by the PFS of the olaparib monotherapy arm, our study population had a poorer prognosis despite our clinical selection of participants with no immediate indication for chemotherapy. Despite baseline characteristics being consistent with prior studies, in this smaller study, we observed a predominance of participants with HRP cancers (58%). We had benchmarked the result of our trial with the ARIEL2 population that appeared, in retrospect, to have a better prognosis. The results of the MEDIOLA trial, more in keeping with our results (compared with the ARIEL2 study) were not available to us at the time of our study planning.

The findings from this trial suggest that treatment with Durvalumab and Olaparib should not replace standard of care, regardless of treatment priming with Olaparib or Cyclophosphamide. As expected, single agent Olaparib was associated with a poor PFS in this study. Surprisingly, there was no significant difference in PFS between HRD and HRP subgroups across all treatment arms.

Authors’ response: We agree with the reviewer’s interpretation of the trial results. With regards to the point of no significant PFS difference between the HRD and HRP

subgroups, a total of 100 (87.7%) out of 114 participants were evaluable for HRD testing, including tissue *BRCA1/2* mutation (sufficient FFPE tumour purity was available for analysis). However, the majority of participants did not undergo repeat biopsy at study entry, as many had small volume disease radiologically. Therefore, the HRD status may not have been reflective of the most current genomic status of the cancer. In our revised manuscript, we have discussed this issue as follows (page 18, paragraph 2):

“Importantly, the utility of the CUP-CC assay may be to characterize the current HRD status of the cancer. In our trial, tissue HRD status of 87.7% participants was based on archival FFPE tissue with adequate tumor purity, where no significant difference in PFS was observed between HRD and HRP subgroups (Figure 4A). Genomic scar tests, such as Myriad CDx, classify the cancer based on both current and prior DNA damage that occurred during tumor evolution, but does not inform about acquired resistance, which could result in a functional change from an HRD to an HRP status, a key factor in resistance to PARPi therapy. For example, in ARIEL2²⁶, differences in the results of LOH assays were reported between pre-treatment biopsies versus archival tumor materials, demonstrating that although 34% of patients with LOH-low cancers based on archival tissue biopsy, were reclassified as LOH-high based on the pre-treatment biopsy, no cases were observed to reverse from LOH-high status to LOH-low status. The conclusion of the ARIEL2 translational studies was that accumulation of genomic scarring is an irreversible process, persisting even as cancers re-acquire functional HRR³⁵. The CUP-CC blood-based assay has the potential to address these challenges.”

The finding of improved PFS for patients who were CUP-CC+ is interesting and warrants further investigation. As all patients in the trial received a Olaparib, it is not yet clear if this test will help to guide patient selection for PARPi – the test certainly appears to have promising prognostic value, but its predictive role remains uncertain.

Authors' response: We agree with the reviewer that the predictive role of CUP-CC has not been established in our study but the findings are very interesting, provocative and certainly deserve further investigation. Work is currently ongoing to definitively establish the predictive value of CUP-CC using samples from maintenance clinical trials of PARPi vs placebo in advanced ovarian cancer. We will report these data in future publications.

Clarity and Context

The abstract could be clearer on several points. The first sentence of the abstract (lines 76-79) only describes two of the three treatment arms (the Olaparib only arm is not included) despite listing the number of patients as 114 (this number includes the 37 patients who received Olaparib alone). Lines 81-83 go on to state that priming with Olaparib or Olaparib + Cyclophosphamide was associated with a 'longer' PFS. It is unclear what the PFS is 'longer' in comparison to. Do the authors mean 'longer'

compared to the single agent PARPi treatment arm (which has still not been mentioned at this point in the abstract)? Or 'longer' than historical PFS results from trials of durvalumab alone? Lastly, the authors have concluded in the abstract that CUP-CC may have the potential to personalize PARPi therapies for BRCA wt patients. As all patients in the trial received a PARPi it is too early to comment if this test will help to guide patient selection, although this will be an area of interest for future study.

Authors' response: We have revised the abstract (page 4) to improve the clarity and also revised the conclusion as follows:

The SOLACE2 trial investigated whether 12-weeks of olaparib, or cyclophosphamide-olaparib priming, improved subsequent impact of durvalumab-olaparib on progression-free survival (PFS), and was superior to olaparib monotherapy without any priming, in participants with platinum-sensitive recurrent ovarian cancer (n=114).

....Priming with olaparib, or cyclophosphamide-olaparib, followed by durvalumab-olaparib, were both associated with longer PFS compared to olaparib monotherapy, but did not reach the pre-specified 36-week trial threshold (PFS36).

....Future studies should investigate whether CUP-CC has the potential to personalize PARP-inhibitor therapies for BRCA wild-type, including HRP patients.

References

As mentioned, a more robust discussion of the current treatment landscape for PSROC would be useful, with particular reference to the uncertain role of PARPi treatment for patients with BRCA1/2 wt status. Reference to other chemotherapy sparing upfront treatments in PROC setting is also needed as discussed.

Authors' response: As outlined in our response above, our revised manuscript has included a new paragraph to discuss the rationale of trial design involving olaparib monotherapy arm, to contextualise our trial to the time of study conception and design, where the evidence to support the role of PARPi as maintenance treatment in PSROC following response to chemotherapy was evolving, and where both HRP and HRD patients appeared to benefit from PARPi, most likely because of intrinsic platinum sensitivity which appeared to be a surrogate for response to a PARPi. We have also expanded our discussion of ARIEL2 data, regarding the role of rucaparib monotherapy and olaparib-bevacizumab from the MEDIOLA trial as examples of chemotherapy-sparing upfront treatments in PSROC.

We have also expanded on the introduction to reference the uncertainty of the role of PARPi treatment in the HRP subpopulation (pages 5-6):

“Poly(ADP ribose) polymerase (PARP) inhibitors, either alone or in combination with bevacizumab, have regulatory approval as maintenance therapy following response to first-line platinum-based chemotherapy. The greatest benefits of PARPi

are observed in patients with either germline or somatic *BRCA* pathogenic variants, and the least benefit in those with homologous recombination proficient (HRP) tumors.

The PRIMA trial of maintenance niraparib in advanced stage high grade serous cancer (HGSOC) recently reported 5-year PFS rates of 22% with niraparib versus 12% with placebo in the intention-to-treat population, with PFS improvement for both HRP and homologous recombination deficient (HRD) subpopulations⁵. However, there was no difference in the median OS between treatment arms, including those with HRD and HRP tumors. The PAOLA-1 trial of first-line maintenance olaparib plus bevacizumab versus bevacizumab also reported no significant difference in OS in the intention-to-treat population, but 5-year OS rates in the HRD population were 65.5% and 48.4% in the olaparib plus bevacizumab and bevacizumab arms respectively⁶. Finally, the addition of durvalumab during both carboplatin-paclitaxel chemotherapy (PC) and maintenance olaparib plus bevacizumab was investigated in the DUO-O trial⁷ and showed a significant PFS prolongation in the *BRCA* wild-type with or without Myriad HRD positive populations, compared to PC-bevacizumab monotherapy, but OS data remain immature and the comparator arm of PC-bevacizumab followed by maintenance bevacizumab plus olaparib was not included, hence preventing certainty around the contribution of durvalumab.

Reviewer #2 (ECR):

Authors' response: We thank reviewer 2 for the contributions to the review process and hope it was an educational experience.

Reviewer #3 (Biostats):

In the SOLACE2 trial (ACTRN12618000686202), 115 patients with platinum-sensitive recurrent ovarian cancer were randomized 1:1:1 to receive olaparib followed by olaparib plus durvalumab (Arm A), olaparib plus low-dose cyclophosphamide followed by olaparib plus durvalumab (Arm B), or olaparib as continuous monotherapy (Arm C). 114 patients were evaluable for analysis. The primary endpoint was 36-week progression-free survival, with a null rate of 47% and target rate of 67%. Both intervention arms had a 36-week progression-free survival rate close to the null rate (Arm A: 47.4%, Arm B: 48.7%), with the authors noting that the noncomparative control arm also had a lower-than-expected 36-week progression-free survival rate (Arm C: 35.1%).

A key correlative objective was to evaluate the prognostic ability of the CUP-CC assay (CCR4 upregulation [CUP] combined with cytokines [IL-6 and IL-8] and chemokines [CCL17 and CCL22]) for patients receiving PARP inhibitor-based treatment, and whether it predicted additional progression-free survival benefit with durvalumab. CUP-CC status could not be determined in 10/114 (8.8%) patients; among the remaining patients, progression-free survival was significantly longer in

CUP-CC+ patients than CUP-CC-patients (HR=0.31, 95% CI [0.19-0.49]), with median progression-free survival of 49.6 weeks for CUP-CC+ patients and 23.4 weeks for CUP-CC- patients. The objective response rate was 58.6% in CUP-CC+ patients and 28.3% in CUP-CC- patients (p=0.002). The CUP-CC assay was not predictive of additional progression-free survival benefit with durvalumab (i.e., Arm A or B versus Arm C). I have the following minor comments.

Authors' response: The reviewer has accurately summarized the key data of our study. We also thank the reviewer for the comments outlined below.

“In conclusion, the SOLACE2 trial shows olaparib-durvalumab and olaparib-LDCy-durvalumab are associated with numerically greater ORR and longer PFS as compared with olaparib monotherapy.” (Lines 424-426). This conclusion seems inconsistent with the study design. Specifically, SOLACE2 was a noncomparative phase II trial (Lines 76 and 436), yet this conclusion focuses on between-arm comparisons (i.e., Arm A versus Arm C, Arm B versus Arm C). As originally designed, for the primary analysis, the Arm A and Arm B 36-week progression free survival rate would each be compared to a null rate of 47%—not to the rate observed in Arm C (which was described as a concurrent noncomparative control arm, Line 498). Please clarify that this conclusion was not based on the intended primary analysis. Please also clarify that the objective response rate and progression-free survival were numerically but not statistically better in Arms A and B versus Arm C.

Authors' response: We confirmed that objective response rate and progression-free survival were numerically, but not statistically better in Arms A and B versus Arm C. We also confirmed that this conclusion was not based on the intended primary analysis of Arms A and B versus Arm C.

In our revised manuscript, we have indicated that (page 19, paragraph 4):

“In conclusion, the SOLACE2 trial showed that neither immune priming strategies were definitively superior to olaparib monotherapy, as the PFS differences were not statistically significant and failed to meet our pre-specified historical threshold of efficacy. Nevertheless, olaparib-durvalumab and olaparib-LDCy-durvalumab were associated with numerically greater ORR and longer PFS as compared with olaparib monotherapy but this study was not powered for relative comparison between treatment arms. Most importantly,....”

“The 36-week PFS rates were 47.4% (95% CI 31.0-62.1), 48.7% (95% 32.5-63.2), and 35.1% (95% CI 20.4-50.3), respectively” (Lines 196-197). The trial's power calculation is based on a one-sided nominal significance level of 0.05, which would be consistent with 90% confidence intervals rather than 95% confidence intervals for these 36-week progression-free survival rates. Please update the confidence intervals or clarify.

Authors' response: The reviewer has correctly indicated that the study was powered based on a one-sided nominal significance level of 0.05. Determination of "success" or "failure" of our trial was based on a 20% improvement from a historical rate of 36-week PFS rate of 47%. In this study, we concluded that we failed to meet this threshold and hence we could not reject the null hypothesis.

We have presented all our data based on the two-sided 95% CI as it is a conventional practice that will enable useful interpretation of the range of plausible values beyond the point estimates.

The arm labels change in the statistical analysis section. Elsewhere in the manuscript Arm C is the control arm (i.e., olaparib as continuous monotherapy) whereas Arm A is the control arm in the statistical analysis section (Lines 496-498). Please update for consistency.

Authors' response: We apologise for the confusion and are grateful for the reviewer picking up the inconsistency. We have revised the manuscript accordingly and have made these corrections (page 22, paragraph 1).

"...For each of the combination treatment arms A and B, 38 participants per arm will be required. Treatment arm C served as a concurrent non-comparative control..."

Lines 176 to 179 state that 114 patients were randomly assigned to Arm A, B, or C whereas the CONSORT diagram shows that 115 patients were randomly assigned. Please clarify in the manuscript text that 115 patients were randomly assigned to Arm A, B, or C but 1 patient was excluded post-randomization due to ineligibility (per the CONSORT diagram).

Authors' response: We apologise for this error which we have addressed. In the revised manuscript (page 8, paragraph 3), we have made this correction to 115 participants:

"From February 2019 to October 2022, 115 participants underwent random assignment, with one ineligible participant excluded post-randomization and did not receive assigned therapy. Participants were assigned to receive olaparib followed by olaparib-durvalumab (arm A, N=38), olaparib-LDCy followed by olaparib-durvalumab (arm B, N=39), or olaparib monotherapy (arm C, N=37)...."

Reviewer #4 (ovarian cancer, therapy/biomarker analysis):

General Feedback:

The SOLACE2 trial presents a comprehensive evaluation of PARPi-based immune priming in recurrent ovarian cancer, integrating translational biomarkers like CUP-CC

to refine patient stratification. The study is methodologically rigorous, with well-defined endpoints and a strong emphasis on correlative analyses to explore immune modulation. The following issues can be addressed to improve the impact of this manuscript.

Authors' response: We thank reviewer 4 for complementing our work and we are also grateful for the points raised to improve our manuscript.

Specific Feedback:

Introduction:

1. The introduction lacks key epidemiological statistics for EOC and PSROC. Including incidence, mortality rates, survival data, and response rates to subsequent chemotherapy lines would provide a clearer clinical context.

Authors' response: We have made the suggested changes in our revised manuscript (page 5, paragraphs 1-2), which are outlined below:

“Epithelial ovarian cancer (EOC) is the global leading cause of death from all gynecological cancers¹. In 2020, there were more than 300,000 new cases globally, and more than 200,000 cancer-related deaths². Most women have advanced stage disease at presentation and are typically treated with surgery, platinum-taxane chemotherapy, and bevacizumab. Although these treatments are associated with high rates of responses, majority will still experience recurrence.

Platinum-sensitive recurrent high grade serous ovarian cancer (PSROC) has conventionally been defined as cancer progression ≥ 6 months after the most recent platinum-based chemotherapy. There is a 50% likelihood of response with retreatment with platinum-based chemotherapy, but there is a declining likelihood of chemotherapy response with each successive line of treatment³. The addition of bevacizumab increases the response rate to 70% and prolongs progression-free survival (PFS)⁴. However, the median overall survival (OS) is still less than 5 years, and similar with chemotherapy alone or addition of bevacizumab.“

2. Bevacizumab and PARP inhibitors are presented as treatment advances, but their limitations are not addressed.

Authors' response: In our revised manuscript (pages 5-6), we have made these changes to reflect differences in the benefit of PARPi according to HRD status and cite that PAOLA-1 did not demonstrate OS benefit with olaparib-bevacizumab as a limitation of this combination.

“Poly(ADP ribose) polymerase (PARP) inhibitors, either alone or in combination with bevacizumab, have regulatory approval as maintenance therapy following response to first-line platinum-based chemotherapy. The greatest benefits of PARPi are observed in patients with either germline or somatic *BRCA* pathogenic variants, and the least benefit in those with homologous recombination proficient (HRP) tumors. The PRIMA trial of maintenance niraparib in advanced stage high grade serous cancer (HGSOC) recently reported 5-year PFS rates of 22% with niraparib versus 12% with placebo in the intention-to-treat population, with PFS improvement for both HRP and homologous recombination deficient (HRD) subpopulations⁵. However, there was no

difference in the median OS between treatment arms, including those with HRD and HRP tumors. The PAOLA-1 trial of first-line maintenance olaparib plus bevacizumab versus bevacizumab also reported no significant difference in OS in the intention-to-treat population, but 5-year OS rates in the HRD population were 65.5% and 48.4% in the olaparib plus bevacizumab and bevacizumab arms respectively⁶. Finally, the addition of durvalumab during both carboplatin-paclitaxel chemotherapy (PC) and maintenance olaparib plus bevacizumab was investigated in the DUO-O trial⁷ and showed a significant PFS prolongation in the *BRCA* wild-type with or without Myriad HRD positive populations, compared to PC-bevacizumab monotherapy, but OS data remain immature and the comparator arm of PC-bevacizumab followed by maintenance bevacizumab plus olaparib was not included, hence preventing certainty around the contribution of durvalumab.

3. The section on CD3+/CD8+ TILs and Tregs is fragmented and not well connected to the study rationale. Clearly explain how Tregs suppress CD8+ TIL function and why overcoming this is key for ICI efficacy. Also, condense this discussion.

Authors' response: We have made the suggested changes in our revised manuscript (page 6, paragraph 2), which are outlined below:

“The impact of immune checkpoint inhibitors (ICI) in women with EOC has been modest^{8, 9}. EOC creates an immunosuppressive tumor microenvironment (TME), by preferentially chemoattracting CCR4-expressing regulatory T-cells (Treg)⁸. Treg in the TME employ multiple mechanisms to suppress CD8 T-cells: via expression of PD-L1 and CTLA-4; secretion of immunosuppressive cytokines IL-10, TGF- β and IL-35; adenosine production by expression of ectoenzymes CD39 and CD73; and other mechanisms¹⁰. Tregs also modulate antigen presenting cells (APCs) in the TME, impairing the ability of APCs to induce and expand CD8 T-cells. The use of ICI to interfere with any single immunosuppressive mechanism employed by Treg, is therefore unlikely to allow sufficient re-engagement of CD8 T-cell effector activity, to result in impressive anti-tumor efficacy. Low dose cyclophosphamide treatment (LDCy) leads to Treg depletion and increased CD8 T cell/Treg ratios in the TME¹¹. We have previously conducted a phase 1 study and showed that the combination of olaparib and LDCy was tolerable and had promising clinical activity, particularly in *BRCA* mutated EOC¹². We therefore hypothesized that LDCy would help remodel the TME, priming the TME for optimal responsiveness to PARPi combination therapies.”

4. The introduction mentions that PARPi activates the STING pathway but does not convincingly explain why combining olaparib with durvalumab is expected to work in ovarian cancer. Acknowledge contradictory evidence, such as the JAVELIN Ovarian 200 trial failure, and explain how LDCy might enhance response.

Authors' response: We have revised this paragraph as follows on pages 6-7:

“PARPi therapy in EOC has been most effective in those with HRD, as DNA damage which arises from single-strand DNA breaks (SSBs) cannot be accurately repaired. In these cancers, PARPi exerts its therapeutic effects through the blockade of DNA damage repair, including of SSBs, leading to the accumulation of toxic DNA double-strand breaks¹³. Of relevance here, PARPi modulate the immune system

through the stimulation of interferon genes (STING) pathway. PARPi induce DNA damage activates the cGAS-STING pathway, increasing IFN- γ release and enhancing T-cell-dendritic cell crosstalk and promoting antigen presentation^{14, 15}. PARPi have synergistic effects with ICI, through DNA damage, cell death, and neoantigen and other antigen release, thereby enhancing ICI and intrinsic immune responses, including enhanced antigen presentation, increasing tumor-infiltrating lymphocytes, upregulation of PD-L1, and reprogramming of other molecules and immune cells involved in the TME^{16, 17}. In this context, it is plausible that prior PARPi alone may prime the TME for subsequent PARPi-ICI therapy. In addition, an increased CD8 T-cell to Treg ratio promoted by LDCy¹⁸, could free the anti-tumor activity of the CD8 T-cells activated by PARPi treatment from Treg immunosuppression.”

The JAVELIN Ovarian 200 was a trial in patients with platinum resistant ovarian cancer and compared chemotherapy plus avelumab vs avelumab monotherapy vs single agent chemotherapy. The JAVELIN Ovarian 200 population is very different to that of SOLACE2 which only included participants with platinum sensitive ovarian cancer. Furthermore, JAVELIN Ovarian 200 did not investigate the use of a PARPi. As outlined in our response in (2), we have revised our manuscript to discuss the large, randomized DUO-O trial, which showed that the addition of durvalumab to chemotherapy plus bevacizumab, followed by the addition of durvalumab to maintenance therapy with bevacizumab plus olaparib, significantly improved PFS (median 45.1 months), as compared to the control arm of chemotherapy plus bevacizumab, followed by bevacizumab maintenance (no olaparib). The median PFS of 45.1 months seen with the durvalumab plus olaparib combination is the longest reported to date in this population of patients, which suggests that the addition of durvalumab may have contributed to this improvement, although the comparator arm of carboplatin-paclitaxel and bevacizumab followed by maintenance bevacizumab plus olaparib was not included, hence preventing certainty around the contribution of durvalumab. The activity of olaparib with durvalumab was also studied in a small phase 2 MEDIOLA trial that demonstrated preliminary efficacy.

5. If CUP-CC is a predictive biomarker, compare it to PD-L1 expression, tumor mutational burden, and IFN- γ signatures. Explain its potential advantage over these established markers.

Authors' response: We have performed additional analyses on PD-L1 expression, tumor mutational burden (TMB) and IFN- γ level. None of these biomarkers were prognostic for PFS. CUP-CC remained a statistically significant prognostic immunological biomarker for PFS.

For PD-L1 expression, we performed OPAL staining for PD-L1 expression on cells within the tumour area (tumour cells and infiltrating lymphocytes). No prognostic signal was seen for PD-L1 expression at a cut off of either 1% or 10%. We have not provided the PD-L1 OPAL data for publication in this manuscript, as we have not yet completed the data analysis for a large number of related immune markers, which

analysis would allow us to ensure that the OPAL dataset is internally consistent. This will be published in a future manuscript.

These figures reporting on PD-L1 staining included below are for reviewers only.

PD-L1 staining was included as part of a 7-colour custom OPAL panel. Processed images were analysed using QuPath where the tumour region and individual cell phenotypes were assigned using manually trained inbuilt machine learning tools. Tumour-stroma boundaries were confirmed by a clinical pathologist using H&E sections. PD-L1 percentage was defined as the proportion of any PD-L1+ tumour cell or infiltrating immune cell within the tumour defined area. The resulting score is equivalent to a combined positive score (<https://doi.org/10.1038/s41379-020-0567-3>) that is generated using IHC, H&E and clinical scoring. Tumor sections from 84 participants were available for scoring.

Kaplan-Meier plots of progression-free survival according to PD-L1 expression based on the cut-off 1%

Kaplan-Meier plots of progression-free survival according to PD-L1 expression based on the cut-off 10%

TMB level was assessed as part of the WES analysis of tumour, estimated as the number of passing non-synonymous somatic variants with a haplotype frequency >20% divided by 36 Mbp to normalize for the length of the analysed region.

In our revised manuscript, we have added on page 14, paragraph 2:

“Data for tumor TMB, categorized based on a cutoff of 5 mutations per megabase (Supplementary Figure 11), and plasma IFN- γ levels, categorized based on the mean distribution (Supplementary Figure 12) are also provided. Neither TMB nor IFN- γ levels were significantly associated with PFS.”

Supplementary Methods:

“TMB was estimated as the number of passing non-synonymous somatic variants with a haplotype frequency >20% divided by 36 Mbp to normalize for the length of the analysed region”

“The cytokines and chemokines were measured from plasma using the Bioplex Multiplex Luminex assays (Bio-rad) following the manufacturer protocol for the premix beads of IL-6 (171-BK29MR2), IL-8 (171-BK31MR2), CCL22 (171-BK41MR2), CCL17 (171-BK53MR2) and IFN γ (171-BK25MR2). The data were acquired and analysed using the BioPlex-200 (Bio-rad). The mean values of the cytokines and chemokines levels from plasma samples of SOLACE2 participants were used to categorize these participants as “high” (above mean) vs “low” (below mean).”

6. Rather than stating that TILs improve survival, explain why their function is impaired in EOC (e.g., high Treg/CD8 ratio, immunosuppressive cytokines, restricted antigen presentation).

Authors' response: We thank the reviewer for this suggestion, and we have included these points in our revised manuscript as outlined in our response in point (3) above.

Results:

7. Discontinuation rates in the priming phase are reported, but baseline characteristics of these patients are not analyzed. Identifying if subgroups (e.g., HRD vs. HRP, CUP-CC+ vs. CUP-CC-) had higher discontinuation rates could offer insights into resistance.

Authors' response: The table below, for the Reviewers, outlined the 21 out of 114 participants who progressed during the priming phase (first 84 days from randomization) by treatment arms and baseline HRD and CUP-CC statuses. With such small numbers, it is not possible to definitively provide any insights into primary resistance to PARPi based therapy.

Participants (n=21) who progressed during the priming phase:

Total evaluable for HRD	Arm A (N=2 of 32)	Arm B (N=3 of 35)	Arm C (N=6 of 33)
HRD	1 (3%)	1 (3%)	1 (3%)
HRP	1 (3%)	2 (6%)	5 (15%)
Total evaluable for CUP-CC	Arm A (N=1 of 35)	Arm B (N=3 of 36)	Arm C (N=6 of 33)
CUP-CC+	0 (0%)	1 (3%)	3 (9%)
CUP-CC-	1 (3%)	2 (6%)	3 (9%)

8. CUP-CC status could not be determined in 8.8% of participants, but the reason is unclear. Clarify whether this was due to sample issues, patient refusal, or technical failures.

Authors' response: All cases of unknown CUP-CC status were due to either sample issues or technical failures, such as insufficient PBMC cells recovered from samples and prevented successful run of the assay. There was no participant refusal.

9. Waterfall plots show RECIST-defined responses, but cases of pseudoprogression or long-term responders not meeting RECIST criteria are not addressed. This is key

for ICI-containing regimens, where pseudoprogression might be misclassified as progression.

Authors' response: In our study, the protocol (submitted together with the manuscript) indicates that if a participant on the durvalumab-containing treatment arm was noted to have progressive disease on a routine CT scan and would like to continue durvalumab treatment on trial, they must undergo a confirmatory CT scan within 4-8 weeks later to confirm progressive disease to evaluate for ORR and PFS according to immune-related RECIST (iRECIST). We have no documented cases of pseudo-progression but acknowledged that not all participants complied with confirmatory CT scan.

10. The report notes that HRP cases with CUP-CC+ had significantly improved PFS compared to CUP-CC-, but it does not compare their outcomes to HRD patients. Clarify if CUP-CC+ can predict response independently of HRD.

Authors' response: In our submitted manuscript, we have provided these comparisons, confirming that CUP-CC can predict response independently of HRD. In Figure 4C, we have shown the following:

HRP & CUP-CC-: 36-week PFS rate: 20.8% (7.6-38.5)

HRP & CUP-CC+: 36-week PFS rate: 64.3% (43.8-78.9)

HRD & CUP-CC-: 36-week PFS rate: 35.7% (13.0-59.4)

HRD & CUP-CC+: 36-week PFS rate: 61.5% (40.3-77.1)

When evaluating CUP-CC status alone without HRD status, the data as reported in Figure 4B are as follows:

CUP-CC-: 36-week PFS rate: 23.9 % (12.9-36.9)

CUP-CC+ 36-week PFS rate: 62.1% (48.3-73.1)

Furthermore, in multivariable analysis (Table 2), CUP-CC remained a significant prognostic factor for PFS even with adjustment of HRD status and other baseline stratification factors.

11. Correlative analysis reports mafosfamide-induced T-cell changes in healthy donors but does not compare findings to patient samples. Such a comparison would validate its clinical relevance.

Authors' response: Mafosfamide induced changes in CCR4 expression in all T cell subsets studied varied greatly both within healthy volunteers, as well as patients with ovarian cancer (Fig Supplementary Figures 2 and 3). However, patients had higher

ratios of CCR4 expressing CD4 T cell/Treg (65.88 vs 20.75, $p < 0.0005$, Welch's t-test) and CCR4 expressing CD8 T cell/Treg ratio (7.714 vs 3.671, $p < 0.0231$, Welch's t-test) in comparison to healthy controls. Together these data indicate cancer independent and cancer related differences contribute to heterogeneity in CUP responses.

Authors' response: We have included this statement as follows on page 21, paragraph 2:

".....This finding should nevertheless be considered in the context of the CUP assay uncovering an individual's overall T cell (beyond Tregs) response to diverse biological signals, resulting in upregulation of CCR4, and promotion of T cell migration into the TME. Higher ratios of effector T cells/Treg in the CUP assay in patients with cancer compared to controls, suggests cancer associated differences in baseline reactivity, supporting the biological utility of the CUP assay in patients. A number of these signals, including histone modifications, can be impacted by PARPi therapy^{33, 34}."

12. Only four patients discontinued due to AEs, which seems low given the reported toxicity. Clarify if dose reductions or treatment delays were needed to manage AEs.

Authors' response: We have provided these data in Table 3 of the submitted manuscript. We have reproduced part of Table 3 below to provide data about treatment interruptions and treatment dose reduction.

	ARM A (N = 38)	ARM B (N = 39)	ARM C (N = 37)
Priming phase			
AE lead to dose interruption - Any grade	2 (5.3%)	5 (12.8%)	2 (5.4%)
AE lead to dose reduction - Any grade	0	0	0
AE lead to permanent discontinuation - Any grade	0 (0.0%)	0 (0.0%)	2 (5.4%)
	ARM A (N = 37)	ARM B (N = 35)	ARM C (N = 30)
Consolidation phase			
AE lead to dose interruption - Any grade	2 (5.4%)	3 (8.6%)	4 (13.3%)
AE lead to dose reduction - Any grade	0	0	0

Discussion:

13. The discussion states that neither intervention arm met the 36-week PFS goal but does not explore biological reasons for the lower-than-expected efficacy.

Authors' response: At the time of designing our trial, we benchmarked the performance of our combination arms A and B against the best available historical control data from ARIEL 2. Our study included predominantly participants with HRP tumors (58%) although this was unplanned as HR status testing was not widely

available during our study design. As outlined by Reviewer 1, our patient population was more like the patients recruited into the phase 2 MEDIOLA trial of olaparib and durvalumab, with and without bevacizumab.

In our revised manuscript, we have outlined the following on page 15, paragraph 2:

“...Priming was tested with either olaparib alone or with the combination of olaparib-LDCy based on the trial primary endpoint of 36-week PFS rate. The choice of this study endpoint was based on our hypothesis that 12 weeks of priming with either strategy, followed by 24 weeks of olaparib-durvalumab maintenance, would demonstrate an improvement in PFS at 36 weeks. However, neither intervention arms A (36-week PFS rate 47.4%) nor B (36-week PFS rate 48.7%) met the threshold of a 36-week PFS rate of 67%. This threshold was based on a 20% improvement from a historical 36-week PFS rate of 47% observed with single agent rucaparib²⁶. Our olaparib monotherapy arm had a worse outcome than expected, with a 36-week PFS rate of 35.1%. Despite baseline characteristics being consistent with prior studies, and our clinical selection of participants with no immediate indication for chemotherapy, in this smaller study, we observed a predominance of participants with HRP cancers (58%).....”

page 16, paragraph 2: “.....At the time of study planning, there was evolving evidence to support the role of maintenance PARPi treatment in PSROC following response to chemotherapy, independent of *BRCA* status as platinum sensitivity was regarded as a surrogate for response to PARPi therapy^{27, 28, 29, 30}. We had also performed a meta-analysis to demonstrate that there was greater PFS benefit with maintenance PARPi in the HRD over the HRP subpopulations, but there was still significant PFS prolongation in the HRP subpopulation when treated with PARPi over placebo³¹. The role of PARPi therapy in the HRP population was not well-defined, and routine genetic testing was still limited to germline *BRCA*, hence our study did not restrict enrolment of study participants at the time. There was also emerging evidence from ARIEL2 study that the reported median PFS with single agent rucaparib was 5.7 months in the loss of heterozygosity (LOH) high subgroup which did not differ significantly from 5.2 months in the LOH low subgroup²⁶. Subsequently, data on the activity of olaparib plus durvalumab was reported in the phase 2 MEDIOLA trial³² that showed ORR of 34.4% and median PFS of 5.5 months in the cohort of *BRCA* wild-type patients treated with olaparib plus durvalumab with PSROC, consistent with the findings from our HRP subpopulation. There is also similarity in the outcomes of the germline *BRCA* mutant cohort of MEDIOLA with our study findings. Both MEDIOLA and our study were recruiting participants at similar times and hence the performance of olaparib plus durvalumab was unknown at our study planning.....”

14. HRD status did not stratify PFS, but potential reasons (e.g., acquired resistance, tumor evolution) are not discussed.

Authors' response: In our trial, tissue HRD status for 100/114 (87.7%) participants, was available from analysis of FFPE archival tissue with adequate tumor purity. Therefore, the HRD status was not reflective of the most current status of the cancer.

In our revised manuscript, we have discussed this issue as follows (page 18, paragraph 2):

“Importantly, the utility of the CUP-CC assay may be to characterize the current HRD status of the cancer. In our trial, tissue HRD status of 87.7% participants was based on archival FFPE tissue with adequate tumor purity where no significant difference in PFS was observed between HRD and HRP subgroups (Figure 4A). Genomic scar tests, such as Myriad CDx, classify the cancer based on both current and prior DNA damage that occurred during tumor evolution, but does not inform about acquired resistance, which could result in a functional change from an HRD to an HRP status, a key factor in resistance to PARPi therapy. For example, in ARIEL2²⁶, differences in the results of LOH assays were reported in pre-treatment biopsies versus archival tumor materials, demonstrating that although 34% of patients with LOH-low cancers based on archival tissue biopsy, were reclassified as LOH-high based on the pre-treatment biopsy, no cases were observed to reverse from LOH-high status to LOH-low status. The conclusion of the ARIEL2 translational studies was that accumulation of genomic scarring is an irreversible process, persisting even as cancers re-acquire functional HRR³⁵. The CUP-CC blood-based assay has the potential to address these challenges.”

15. ORR differences between treatment arms are noted but lack interpretation. The 60% ORR for olaparib/LDCy/durvalumab is much higher than the 38.5% for olaparib/durvalumab—does this suggest immune priming or just expected variability?

Authors' response: In our HRD subpopulation, we have reported ORR for olaparib/LDCy/durvalumab as 60.0% (9/15) vs olaparib/durvalumab (5/13, ORR=38.5%) and olaparib monotherapy (5/14, ORR=35.7%). Within our BRCA mutant cohort (n=13), ORR for olaparib/LDCy/durvalumab was 100% (5/5) vs olaparib/durvalumab (3/3, ORR=100%) and olaparib monotherapy (4/5, ORR=80.0%). When we reanalysed the data to evaluate all BRCA wild-type as a single cohort (n=101), ORR for olaparib/LDCy/durvalumab was 47.1% (16/34) vs olaparib/durvalumab (13/35, ORR=37.1%) and olaparib monotherapy (9/32, ORR=28.1%).

When compared with MEDIOLA trial, ORR was 92.2% in the germline BRCA mutant cohort (N=51), and ORR was 34.4% in the non-germline BRCA mutant cohort (N=32), in patients treated with olaparib/durvalumab.

Numerically, our study participants treated with olaparib/LDCy/durvalumab have the highest ORR, and appear to outperform the MEDIOLA trial population. However, the patient numbers are small and we did not use ORR as the primary trial endpoint. As indicated above, the MEDIOLA trial data were not available at the time of our study planning. We therefore are unable to conclude definitively the superiority of olaparib/LDCy/durvalumab.

16. CUP-CC was not predictive of additional benefit from LDCy or durvalumab, but the discussion does not speculate on why immune priming failed to improve efficacy.

Authors' response: Our small phase 2 trial was designed to be non-comparative and was underpowered for analysis to demonstrate predictive value of CUP-CC for the additional benefit from LDCy or durvalumab over olaparib monotherapy.

We have revised our manuscript as outlined in our response to point (13). Our trial had a study population with predominantly HRP tumors, and we had benchmarked the result of our trial with the ARIEL2 population that appeared to have a better prognosis. Data from MEDIOLA trial was not available to us at the time of our study planning which would have enabled us to better calibrate the results of our study.

17. The claim that CUP-CC could expand PARPi use in HRP patients is interesting but lacks comparative data with existing biomarkers like PD-L1 expression and TMB. This would clarify its clinical utility.

Authors' response: Please see our response to point (5) above. Please also see the additional data on PD-L1 expression with figures produced for reviewers as outlined point (5) above.

Data on tumor mutational burden (Supplementary Figure 11) and plasma IFN- γ levels (Supplementary Figure 12) are now provided in the revised manuscript.

18. HRD and HRP subgroups showed no significant PFS difference, yet the discussion does not explore why. Could high rates of HRD reversion mutations or epigenetic changes restoring homologous recombination proficiency be involved?

Authors' response: Please see our response above in point (14).

19. CUP-CC is described as a possible replacement for HRD genomic tests, but no direct evidence supports this. Can CUP-CC track treatment-induced changes in the tumor microenvironment over time?

Authors' response: We have stated in our submitted manuscript that the utility of CUP-CC based on SOLACE2 will need validation in other clinical trial datasets and also prospective testing before replacement for HRD genomic tests. We are currently still testing our longitudinal samples to track changes in CUP-CC status over time. This work is ongoing and will be reported in future publications.

20. Was prior PARP inhibitor (PARPi) exposure an exclusion criterion, *as prior exposure could influence response to Olaparib?*

Authors' response: Prior exposure to olaparib or any PARPi is a study exclusion criterion. In the revised manuscript (page 20, paragraph 2), we have stated the following:

"...Participants were excluded if they had prior use of a PARPi, active autoimmune or inflammatory disorders, prior diagnosis of myelodysplastic syndrome or acute myeloid leukaemia. The full eligibility criteria are available in the study protocol (supplementary)."

21. *Why CA-125 (<100 vs. ≥100 U/mL) was used as a stratification factor over standard markers like treatment-free interval or platinum resistance scores?*

Authors' response: In our submitted manuscript, we indicated that the SOLACE2 trial was stratified by site/institution, platinum-free interval (6-12 months vs >12 months), CA-125 (<100 U/mL vs ≥ 100 U/mL), measurable disease (present vs absent) and germline BRCA mutation status (present vs absent).

We have therefore stratified based on platinum-free treatment-free interval based on conventional cutoff of 6-12 months vs >12 months. The information is displayed in page 20, paragraph 3.

22. *Whether the trial used blinded assessment for tumor response evaluations, as blinding is crucial to reduce bias in non-comparative trials?*

Authors' response: Our study was open-label, with no blinded assessment for clinical tumor response evaluations. We have acknowledged this as a limitation in the revised manuscript (pages 15-16):

".....Despite baseline characteristics being consistent with prior studies, and our clinical selection of participants with no immediate indication for chemotherapy, in this smaller study, we observed a predominance of participants with HRP cancers (58%). Blinded evaluations for tumor response were also not performed which might also bias the results. However, correlative data analysis (immune biomarkers and tumor genetics) was performed blinded to treatment group and clinical outcome status."

Our revised manuscript has also been further edited to improve on clarity and to comply with the word limits of Nature Communications article. We have also added Ms Imalki Kariyawasam as an additional author whose work was required for the immune analyses requested by Reviewer 4.

We hope that we have satisfactorily addressed all comments and questions of the reviewers, and we look forward to acceptance of this manuscript for publication.

Thank you.

With kind regards,

Prof Chee Khoon Lee on behalf of all authors